

# Discriminating between "Drizzle or rain" and sea salt aerosols in Cloudnet for measurements over the Barbados Cloud Observatory

Johanna Roschke[1], Jonas Witthuhn[1,2], Marcus Klingebiel[1], Moritz Haarig[2], Andreas Foth[1], Anton Kötsche[1], and Heike Kalesse-Los[1]

[1]Leipzig Institute for Meteorology (LIM), Leipzig University, Leipzig, Germany
[2]Leibniz Institute for Tropospheric Research (TROPOS), Leipzig, Germany

**Correspondence:** heike.kalesse-los@uni-leipzig.de

**Abstract.** The highly sensitive Ka-band cloud radar at the Barbados Cloud Observatory (BCO) frequently reveals radar reflectivities below -50 dBZ within updrafts and below the cloud base of cumulus clouds. These so-called haze echoes are signals from hygroscopically grown sea salt aerosols. Within the Cloudnet target classification scheme, haze echoes are generally misclassified as precipitation ("Drizzle or rain"). We present a technique to discriminate between "Drizzle or rain" and sea salt aerosols in Cloudnet that is applicable to marine Cloudnet sites. The method is based on deriving heuristic probability functions utilizing a combination of cloud radar reflectivity factor, radar mean Doppler velocity and ceilometer attenuated backscatter coefficient. The method is crucial for investigating the occurrence of precipitation and significantly improves the Cloudnet target classification scheme for the measurements over the BCO. The results are validated against the amount of precipitation detected by the Virga-Sniffer tool. We analyze data for the measurements in the vicinity of the BCO covering two years (July 2021 - July 2023) as well as during the ElUcidating the RolE of Cloud–Circulation Coupling in ClimAte (EUREC[4]A) field experiment that took place in Jan–Feb 2020. A first-ever statistical analysis of the Cloudnet target classification product including the new "haze echo" target over two years at the BCO is presented. In the atmospheric column above the BCO, "Drizzle or rain" is on average more frequent during the dry season compared to the wet season, due to the higher occurrence of warm clouds contributing to the amount of precipitation. Haze echoes are identified about four times more often during the dry season compared to the wet season. The frequency of occurrence of "Drizzle or rain" in Cloudnet caused by misclassified haze echoes is overestimated by up to 16 %. Supported by the Cloudnet statistics and the results obtained from the Virga-Sniffer tool, 48 % of detected warm clouds in the dry and wet season precipitate. The proportion of precipitation evaporating fully before reaching the ground (virga) is higher during the dry season. During EUREC[4]A, precipitation from warm clouds was found to reach the ground more frequently over the RV *Meteor* compared to the BCO.

## 1 Introduction

The trade wind regions across the tropical oceans are characterized by shallow cumulus clouds, or in other words "trade wind cumuli", which exhibit various forms and structures (Medeiros et al., 2015; Stevens et al., 2016, 2020). Trade wind cumuli redistribute moisture within the atmosphere, necessary for deep convection and help drive the Hadley cell circulation (Lonitz,





2014). Their spatial structure and evolution can be influenced by precipitation. Precipitation from trade wind cumuli often
occurs in the form of drizzle (Wu et al., 2017). Accurate estimates of the frequency and intensity of this type of precipitation,
especially over extended periods, are rare.

A suitable location with the necessary instrumentation to study precipitation from trade wind cumuli is the Barbados Cloud
Observatory (BCO). The observatory is equipped with ground-based remote sensing instruments and provides consistent long-
term observational data sets that make it a unique measurement station in the maritime tropics (Stevens et al., 2016). The highly
sensitive Ka-band cloud radar at the BCO frequently reveals radar reflectivities between -50 to -65 dBZ, that are referred to
as haze echoes (Klingebiel et al., 2019). The majority of these haze echoes occur within updrafts and below the cloud base of
shallow cumulus clouds and are caused by hygroscopically grown sea salt aerosols (Klingebiel et al., 2019).

The processes leading to the frequent existence of large sea salt aerosols in the subcloud layer at BCO are illustrated in Fig. 1
and can be summarized as follows: Sea salt aerosols are injected into the atmosphere from white caps of breaking waves. White
cap formation strongly depends on wind speed which controls the concentration and sizes of sea salt aerosols (Woodcock, 1953;
Lewis and Schwartz, 2004). For wind speeds larger than $5\,\mathrm{m\,s^{-1}}$ the percentage covered by whitecaps increases as the square
of the wind speed. While wind speed is certainly the most important factor, parameters such as sea surface temperature and
salinity, atmospheric stability, or wave height have been found to influence the production of sea salt aerosols at the ocean
surface (Lewis and Schwartz, 2004). Over Barbados, the average near-surface relative humidity is found to be larger than $70\,\%$
throughout the year (Klingebiel et al., 2019). From the surface, the relative humidity increases almost linearly to saturation
at the base of the cloud (Nuijens et al., 2015). Consequently, the humidity conditions over Barbados favor sea salt particles
to exist in a deliquescent state (Haarig et al., 2017). Within the convective boundary layer, sea salt aerosols are transported
towards higher altitudes (indicated by positive radar mean Doppler velocities in Fig. 1). Due to their hygroscopicity, the sea
salt particles grow in size with increasing relative humidity which can be observed by an increase in the radar reflectivity factor
from lower subcloud layer altitudes towards the cloud base (Klingebiel et al., 2019). The diameter of sea salt aerosols can range
between $0.2\,\mathrm{\mu m}$ to greater than $50\,\mathrm{\mu m}$ (Lewis and Schwartz, 2004). Measurements of marine aerosol particles in Oahu, Hawaii,
show, that coastal breaking waves produce aerosol particles with dry diameters of around $7\,\mathrm{\mu m}$ that double in size for ambient
relative humidities of $80\,\%$ (Clarke et al., 2003). Klingebiel et al. (2019) retrieved equivolumetric particle diameters following
the method of O'Connor et al. (2005) for atmospheric profiles containing haze echoes at the BCO. The retrieved diameters
range between 6 and $11\,\mathrm{\mu m}$ with total number concentrations of $20\,\mathrm{cm^{-3}}$ near the cloud base (Klingebiel et al., 2019). This
means that sea salt aerosols can grow to the size of cloud droplets, which typically have a diameter of around 5 to $20\,\mathrm{\mu m}$ in
maritime clouds (Wallace and Hobbs, 2006). The size range in which the transition between cloud droplets and precipitation
particles happens varies in the literature. Drizzle can be defined as drops that are large enough to have fall velocities that exceed
the typical fluctuations of vertical velocity in the cloud. For a reasonable range of stratocumulus vertical velocities of 0.1 to
$1\,\mathrm{m\,s^{-1}}$, the corresponding diameters are approximately 50 to $250\,\mathrm{\mu m}$ (Glienke et al., 2017). This is also in line with O'Connor
et al. (2005), who defined drizzle as drops with a diameter greater than $50\,\mathrm{\mu m}$.

Various threshold-based techniques exist to distinguish between clouds, drizzle, and sea salt particles from ground-based
cloud radar observations. However, the proportion of radar reflectivity signals associated with cloud droplets, precipitating



hydrometeors, or sea salt aerosol particles is poorly constrained and often depends on the choice of threshold values. Nuijens
et al. (2014) e.g., chose a threshold value of -40 dBZ for the radar reflectivity measurements to filter out clutter and atmospheric plankton from the measurements of the Ka-band cloud radar (KATRIN) at the BCO. Their study demonstrates, that the proportion of detected clouds near the lifting condensation level (LCL) increases when lowering the radar reflectivity threshold as more optically thin clouds will contribute to the derived cloud cover.

Lamer et al. (2015) also commented on the challenges in choosing an appropriate method to filter out cloud radar echoes
affected by precipitation size particles and sea clutter at BCO. They discarded the lowest 510 m of KATRIN radar data which does not eliminate cloud echoes but rain shafts below the cloud base. Acquistapace et al. (2019) introduced a new algorithm to detect drizzle development in warm clouds based on the skewness of the Ka-band radar Doppler spectrum. For a case study over the BCO in 2013, they found that radar reflectivities below -50 dBZ exist when the algorithm identifies precipitating particles or Cloudnet classifies "Drizzle or rain". With the replacement of the KATRIN cloud radar by the CORAL cloud radar in April
2015, studies commonly employed a threshold of -50 dBZ, which is commonly used across studies, to filter out haze echoes in the BCO measurements (Klingebiel et al., 2019; Schulz et al., 2021; Vogel et al., 2021). However, it remains unclear if this threshold also eliminates radar echoes from precipitation below the cloud base at BCO as the Cloudnet target classification suggests. If yes, radar-based precipitation statistics would be influenced.

Synergistic retrievals such as Cloudnet, provide the potential to identify different hydrometeors by applying state-of-the-
art data processing chains for a complex combination of data from ground-based remote sensing instruments (Illingworth et al., 2007). Cloudnet offers a range of products, including the target classification scheme, designed to identify the physical phase of hydrometeors. However, based on its simple approach of classifying continuous radar reflectivities below cloud base as precipitation, haze echoes from sea salt aerosol at BCO are classified as "Drizzle or rain" within the Cloudnet target classification scheme.

More recently, Kalesse-Los et al. (2023) introduced the functionality of a tool, called the Virga-Sniffer, to distinguish precipitation that fully evaporates in dry subcloud layers (virga) from precipitation that reaches the ground based on a synergy of cloud radar and ceilometer measurements. During the Elucidating the Role of Cloud–Circulation Coupling in Climate (EUREC[4]A) field experiment (Stevens et al., 2021) in January and February 2020 in the tropical western Atlantic, they observed that a substantial amount of trade wind cumuli over the research vessel (RV) *Meteor* produce virga (Kalesse-Los et al., 2023).

In this study, we demonstrate a technique to discriminate between the precipitation class "Drizzle or rain" in Cloudnet and sea salt aerosols that was developed for BCO and is applicable to observations from other marine Cloudnet sites that also detect these haze echoes. The results are validated against the amount of precipitation detected by the Virga-Sniffer tool. For the first time, we conduct statistics from the Cloudnet target classification product including the proposed "Haze echo" class and the amount of virga detected by the Virga-Sniffer tool for long-term measurements over the BCO.

Specifically, the structure of the paper is as follows: The relevant instrumentation, data sets and tools, including the standard Cloudnet target classification and the Virga-Sniffer tool, are described briefly in Sect. 2. The method developed in the present study for discriminating between "Drizzle or rain" and sea salt aerosols (from here on used synonymously for "haze echoes"), as well as the object-based cloud type classification type technique is introduced in Sect. 3. For the long-term statistics based



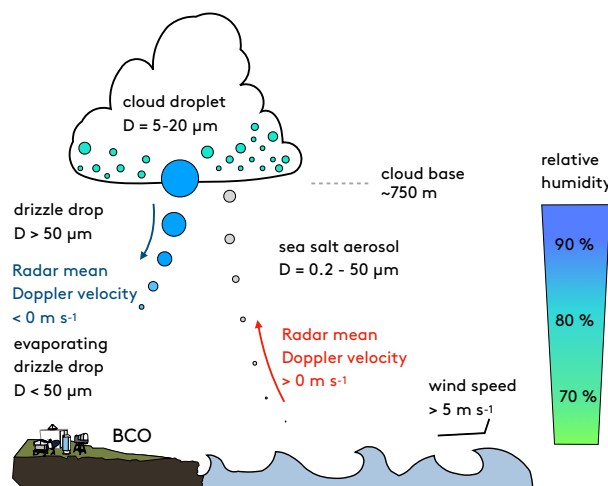

**Figure 1.** Schematic showing the range of diameters (D) for different hydrometeors and sea salt aerosols occurring in the marine boundary layer over the Barbados Cloud Observatory (BCO).

on two years of observations between July 2021 and July 2023 at the BCO, a seasonal comparison of the results obtained by
Cloudnet and the Virga-Sniffer tool at the BCO is included in Sect. 4, the ElUcidating the RolE of Cloud–Circulation Coupling in ClimAte (EUREC[4]A) campaign period from January to February 2020 is also analyzed in detail. Moreover, the impact of environmental factors controlling the occurrence of haze echoes is investigated. To validate the results of the proposed modified Cloudnet target classification including haze echoes, the results are compared to the Virga-Sniffer tool in Sect. 4.3. Conclusions are given in Sect. 5.

**2   Instruments, data sets, and tools**

Continuous observations at the BCO started in April 2010. Core instruments include a cloud radar, a multiwavelength polarization Raman lidar, a ceilometer, a microwave radiometer (MWR), and a micro-rain radar (MRR) (Stevens et al., 2016). Such advanced ground-based instrumentation forms the basis of Cloudnet (Illingworth et al., 2007). A summary of BCO instrumentation used in this study is provided in the following and in Table 1.

**2.1   Cloud radar**

The polarized Ka-band Doppler cloud radar operates at a frequency of $35\,\mathrm{GHz}$ and is part of the Combined Radar And Lidar instrument (CORAL) at BCO. The CORAL radar has been operating since April 2015 as a successor of the lower-sensitivity cloud radar KATRIN which had been in operation since December 2010. Until June 2018, measurements with the CORAL radar were obtained at a temporal resolution of $10\,\mathrm{s}$, from July 2018 onward the temporal resolution was increased to $2\,\mathrm{s}$. The
calibration of the radar shows an uncertainty of $1.3\,\mathrm{dB}$ for the radar reflectivity measurements (Görsdorf et al., 2015). In the





employed measurement configuration the radar has a sensitivity of -70 dBZ at an altitude of 500 m and -48 dBZ at an altitude of 5 km. It measures in a vertical range between 150 m and 18.9 km with a range resolution of 31.18 m. The large antenna generates a narrow beam width of $0.3\,°$. The Doppler resolution is less than $0.02\,\mathrm{m\,s^{-1}}$ and the Nyquist velocity is $\pm 10\,\mathrm{m\,s^{-1}}$ (Klingebiel et al., 2019).

## 2.2 Ceilometer

The Jenoptik CHM15k laser ceilometer at the BCO operates at a wavelength of 1064 nm. Backscattered energy from aerosols and clouds is measured with a temporal resolution of 10 s up to 15 km height and a range resolution of 15 m (Heese et al., 2010). The ceilometer is well-suited for identifying aerosol- and cloud liquid layers in vertical profiles. Its ability to detect liquid layers is however often limited to the lowest cloud layer (in multi-layer cloud situations) or to the cloud base due to strong signal attenuation. Additionally, strong signal attenuation in heavy precipitation can hamper the cloud base detection (Tuononen et al., 2019).

### 2.2.1 Micro rain radar

The micro rain radar (MRR) operating at the BCO is a compact frequency-modulated continuous wave (FMCW) vertically pointing Doppler radar. The instruments measure at a frequency of 24.23 GHz up to 3.1 km above the ground level. A summary of technical features of the MRR is given in Table 1. The temporal resolution is 60 s and the range gate resolution is 100 m. Quantities like rain rates, drop size distributions, radar reflectivity, fall velocity of hydrometeors and other rain parameters can be retrieved simultaneously (Peters et al., 2005).

### 2.2.2 Microwave radiometer

The scanning Radiometer Physics HATPRO radiometer (SUNHAT) has two receivers. It measures seven brightness temperatures around the water vapor absorption band between $22-31$ GHz and in the oxygen absorption complex between $51-58$ GHz. Measurements around the water vapor absorption line are used to derive a column-integrated liquid water path (LWP). The vertical resolution is less than 40 m in the sub-cloud layer with a temporal resolution of 4 s. Data from the current microwave radiometer are available since April 2017 (Stevens et al., 2016).

## 2.3 The synergistic retrieval Cloudnet

The European research project Cloudnet started in 2002 to optimize the representation of clouds in forecast models. From a combination of remote sensing instruments, uniform data sets in the NetCDF format are created within the Cloudnet processing scheme (http://Cloudnet.fmi.fi). On this basis, algorithms for the evaluation of cloud profiles can be applied for various measurement stations (Illingworth et al., 2007). To reliably process gigabytes of cloud remote sensing data per day in near real-time, the CloudnetPy software package was designed. Compared to the older Cloudnet Matlab-software, the new Python implementation of the Cloudnet processing scheme is open source, more user-friendly, and contains detailed documentation



**Table 1.** Specifications of the instruments from the Barbados Cloud Observatory and measured quantities used in Cloudnet

| Instrument (Reference) | Frequency $f$ Wavelength $\lambda$ | Measured quantities relevant in Cloudnet | Temporal resolution | Vertical range | Vertical resolution |
|---|---|---|---|---|---|
| Doppler cloud radar (CORAL) METEK MIRA-35 (Görsdorf et al., 2015) | $f = 35.5\,\mathrm{GHz}$ $8.2\,\mathrm{mm}$ | Radar reflectivity factor Doppler velocity Spectral width Linear depolarization ratio | 2 s | 156 m - 18.98 km | 31.18 m |
| Ceilometer Jenoptik CHM15k (Heese et al., 2010) | $\lambda = 1064\,\mathrm{nm}$ | Attenuated backscatter coefficient | 10 s | 40 m - 15.37 km | 14.99 m |
| Micro-rain radar METEK MRR-2 (Peters et al., 2005) | $f = 24\,\mathrm{GHz}$ | Rain rate | 60 s | 125 m - 3.13 km | 100 m |
| Microwave radiometer RPG-HATPRO-G2 (Rose et al., 2005) | $f = 22.24 - 31.4\,\mathrm{GHz}$ $f = 51.0 - 58\,\mathrm{GHz}$ | Liquid water path | 4 s | - | - |

and tests. Thus, CloudnetPy enables the research community to further develop methods for new products (Tukiainen et al., 2020).

The basic instrumentation of a Cloudnet station includes a cloud radar and a ceilometer. Additional instruments are a MWR, a MRR and a rain gauge. The MWR is needed to provide liquid water path, although W-band radars with a passive channel as
described in Küchler et al. (2017) can be used at this stage for LWP determination making the MWR an optional instrument. The observations from the instruments are combined with thermodynamic data from a model or radiosonde to accurately characterize clouds up to 15 km with high temporal and vertical resolution (Illingworth et al., 2007). In this study, Cloudnet data is processed with CloudnetPy (Tukiainen et al., 2020, version 1.43.1).

The Cloudnet target classification is a profile-based Cloudnet data product that provides information about atmospheric
constituents in the atmospheric column above the observation site. For each time step, the cloud base height is determined to identify which grid points are part of a cloud. The cloud base height (CBH) is determined by analyzing the shape of the attenuated backscatter profile from the ceilometer data using the method of Tuononen et al. (2019). Due to its sensitivity to particle number concentration, the ceilometer backscatter signal strongly increases at cloud base. Further up, the ceilometer signal quickly attenuates while penetrating the cloud liquid layer. Accordingly, the CBH is detected from the ceilometer at-
tenuated backscatter coefficient profiles that show a sudden increase and decrease (in CloudnetPy within 150 m around the maximum value of attenuated backscatter coefficient). Profiles that contain precipitation also lead to increased values in the attenuated backscatter coefficient. However, due to the lower number concentration of precipitation particles, the attenuation of the ceilometer signal is less strong compared to when the signal reaches the liquid cloud base. A criterion that determines the



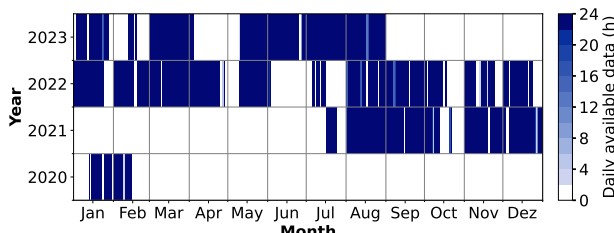

**Figure 2.** Daily available data of the Cloudnet target classification product and data produced with the Virga-Sniffer tool over the BCO from 2020 to 2023.

vertical width of the peak in the attenuated backscatter coefficient profile therefore prevents the cloud base from being detected
below the true cloud base in situations where precipitation is present (Tuononen et al., 2019).

Cloud top height (CTH) in Cloudnet can be determined using the ceilometer based on a similar method as the CBH detection. However, the radar is used for CTH detection when it detects a signal above that defined by the ceilometer which is the case for all but very optically thin clouds. Consequently, CTH is usually taken to be the last pixel below the pixel where no radar signal is detected (Hogan and O'Connor, 2004). A pixel represents the smallest unit of measurement in both time and space
within the Cloudnet dataset.

In the Cloudnet target classification, grid points are classified as "Cloud droplets only" when the wet-bulb temperature $T_\mathrm{w}$ is above zero and no falling particles are identified between CBH and CTH. When drizzle is present inside the cloud, the whole profile between CBH and CTH is classified as "Drizzle/rain & cloud droplets". For example, drizzle inside a cloud is identified when precipitation falls from a higher layer into the cloud or when the radar reflectivity factor increases towards the cloud base
due to the coalescence of drops. "Drizzle or rain" is classified for all radar signals below cloud base when the surface rain flag indicates that precipitation is reaching the ground. When the rain flag is not set, radar signals are only classified as "Drizzle or rain" when they are continuous below CBH. Rain reaching the surface flag is set when the radar reflectivity factor exceeds $0\,\mathrm{dBZ}$ at the lowest radar range gate (here: $156\,\mathrm{m}$) or when additional instruments like the MRR indicate that precipitation is reaching the ground (Hogan and O'Connor, 2004). The interested reader is referred to the original paper by Hogan and
O'Connor (2004) for additional information about the Cloudnet target classification procedure.

An overview of the data availability of the Cloudnet target classification product from 2020 to 2023 is presented in Fig. 2. Cloudnet data after August 2023 was not processed for this study. Instruments were not operating at the BCO after the EUREC[4]A campaign in February 2020 until July 2021. Data between 2015 and 2020 was not included in the study because of frequent issues with the time stamp of the ceilometer data which hampered Cloudnet processing.

**2.4   Virga-Sniffer tool**

The Virga-Sniffer is a Python package tool developed by Kalesse-Los et al. (2023) that serves as a profile-based detection scheme for identifying precipitation, virga, and clouds using observations of vertically pointing cloud radar reflectivity and





ceilometer measurements of CBH. The detection process relies on a set of empirical thresholds, which have been manually adjusted for this study using the Cloudnet data from the BCO. The configuration of the Virga-Sniffer tool employed in this study can be found in Table A1 (configuration 1) in Appendix A. The cloud radar data is fundamental to the Virga-Sniffer as it establishes the temporal and vertical resolution for the detection algorithm. Specifically, for the BCO dataset, CORAL cloud radar data were taken from the Cloudnet target categorization dataset with a temporal and vertical resolution of $30\,\mathrm{s}$ and $30\,\mathrm{m}$, respectively. As further Virga-Sniffer input, CBH is taken from the ceilometer internal cloud base detection algorithm. In this study, the CBH data from the Cloudnet target classification product is used to fill gaps in the ceilometer CBH data. In multi-layer cloud situations, CBH is assigned to specific layers over the processing interval by more than the set threshold of $1000\,\mathrm{m}$ (`cbh_layer_thres`; see Table A1 in Appendix A). Individually configurable modular methods can be applied to the input data in CBH pre-processing. In general, the modules enable sorting CBHs into the corresponding layers, creating or merging new layers. Moreover, CBHs within a layer are smoothed over a certain period (defined by `cbh_smooth_window`). In addition, the lowest CBH can be replaced by the LCL, but this is prevented in this study by setting `lcl_replace_cbh = False`. A detailed explanation of CBH processing can be found in Kalesse-Los et al. (2023).

The detection of precipitation, clouds, and CTH is performed by analyzing radar signals. Precipitation is identified at each range gate of the radar reflectivity mask below CBH. The process involves downward assignment from the CBH until the lowest radar signal. The downward assignment is continued until the gap larger than the specified threshold of $150\,\mathrm{m}$ is encountered. This ensures that tilted fall streaks and vertically separated cloud layers are captured. Radar-based surface rain flag is set if $Z_\mathrm{e}$ is larger than $0\,\mathrm{dBZ}$ in the first radar rage gate. Additionally, surface rain information is obtained from the rain flag in the Cloudnet data, which incorporates measurements of the MRR. The virga-mask is further refined through the incorporation of cloud radar mean Doppler velocity data (see Fig. 3). Firstly, virga are restricted to falling hydrometeors (negative mean Doppler velocities via the velocity mask (`mask_Vm`). Secondly, data points that satisfy the following condition are labeled as clutter:

$$V_\mathrm{m} < -m \cdot (Z_\mathrm{e}/67(\mathrm{dBZ})) - c \qquad (1)$$

with $m = -8$ and $c = -6$. The radar reflectivity factor $Z_\mathrm{e}$ is scaled by $67\,\mathrm{dBZ}$, this represents the minimum reflectivity measured by the CORAL cloud radar during the time from 1 July 2021 until 1 July 2022. Figure 3, displays the 2D-Histogram of the radar reflectivity factor and mean Doppler velocity for observations at the BCO during that period. To filter out sea salt aerosols (radar reflectivities below -50 dBZ) within updrafts, virga events are restricted to the occurrence of negative Doppler velocities. The refinement of the clutter mask is achieved through the parameter values, `clutter_m` and `clutter_c`. This ensures that low radar reflectivities are categorized as clutter, even in the presence of negative Doppler velocities. However, it is important to consider that evaporating drizzle drops may exhibit radar reflectivities comparable to those of sea salt aerosols in downdrafts. As a result, the clutter mask is defined in a balanced manner, capturing both haze echos in downdrafts and evaporating drizzle instances when Doppler velocities are negative. Thus, the Virga-Sniffer can also be utilized to determine haze echoes. Here, haze echoes identified by the Virga-Sniffer include all remaining unclassified radar signals below CBH with $Z_\mathrm{e} < $-50 dBZ and with positive mean Doppler velocities (updrafts). Note that haze echoes have not been observed by





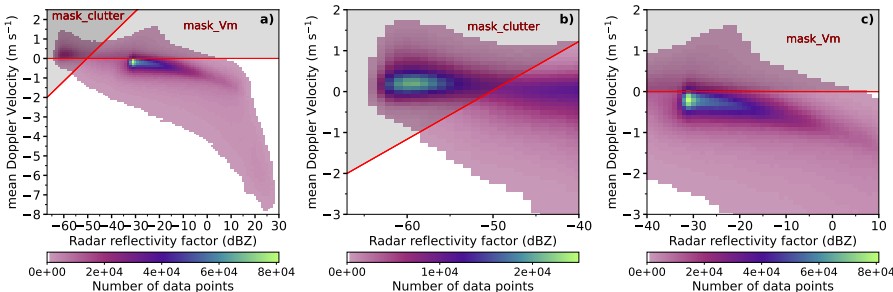

**Figure 3.** BCO 35 GHz cloud radar (1 July 2021 – 1 July 2022): Histograms of the mean Doppler velocity $V_\mathrm{m}$ and radar reflectivity factor. Data points from the grey-shaded areas are not considered as virga. The velocity mask (`mask_Vm`) restricts virga events to falling hydrometeors. The choice of the clutter mask (`mask_clutter`) determines the number of pixels classified as haze echos. Radar reflectivity bin width is 1 dBZ and the Doppler velocity bin width 0.1 m s$^{-1}$. Panel (b) and panel (c) represent the same histogram as in panel (a) but zoomed in for different intervals of the radar reflectivity factor.

Kalesse-Los et al. (2023) for the measurements over the RV *Meteor* due to the lower sensitivity of the 94 GHz which has a minimum valid reflectivity value of -60 dBZ.

An example of the Virga-Sniffer output can be seen in Fig. 4 (b) together with the Cloudnet target classification (a) for BCO observations on 2 December 2021. Note that there is a large proportion of pixels misclassified as "Drizzle or rain" by Cloudnet below shallow cumulus clouds after 07:15 UTC that are mostly unclassified radar signals with radar reflectivity factor below -50 dBZ in the Virga-Sniffer output. Also note that for this case study the CBH of the Virga-Sniffer tool differs from the CBH in Cloudnet (Fig. 4). In this case, the differences in CBH are due to the configurations of the Virga-Sniffer, where the height of the LCL is used as a replacement for the lowest CBH. The LCL is calculated from surface measurements of pressure, temperature and humidity and added as the lowest potential CBH layer for the CBH processing. The LCL is added but does not automatically replace the lowest CBH layer (`LCL_replace_cbh` is not used). Instead, the calculated LCL values are used to fill gaps of the lowest CBH layer. In the CBH processing, CBH layer with similar altitudes (within the threshold `cbh_layer_thres` of 1000 m in the Virga-Sniffer configuration used for BCO data processing) are merged by using their mean altitude value. Reducing the threshold value of `cbh_layer_thres` to a lower value would result in a high number of CBH layers if the ceilometer CBH fluctuates strongly.

## 3 Methodology

This section gives an overview of the method that was developed to discriminate between sea salt aerosols and "Drizzle or rain" in Cloudnet. The method is similar to the approach for insect detection in Cloudnet, which ensures that it can be implemented within the Cloudnet target classification scheme and that it is configurable for marine Cloudnet sites and their particular instrumentation.





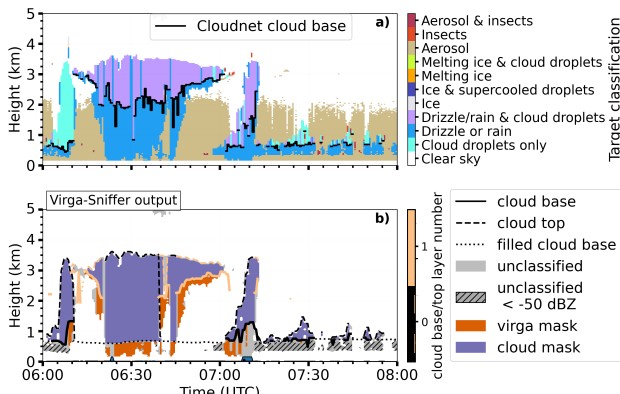

**Figure 4.** BCO case study of 2 December 2021: Cloudnet target classification in panel (a) and Virga-Sniffer output in panel (b). The filled cloud base refers to time steps where the LCL is used as the CBH. Haze echoes from the Virga-Sniffer output are unclassified, with radar reflectivities lower than -50 dBZ. Rain reaching the surface is marked in blue along the x-axis.

## 3.1 Defining a sea salt (or haze echo) probability in Cloudnet

Haze echos are classified by estimating the heuristic probability from individual parameters (namely radar reflectivity, radar mean Doppler velocity and ceilometer attenuated backscatter coefficient) and combining these probabilities. For each observation, a probability array is estimated. The probability of an observed value being less than or equal to a given input value is estimated with the cumulative distribution function of the normal distribution of the input variable. The probability function ($P(X)$) in Eq. (2) is a built-in function in SciPy called `scipy.stats.norm.cdf` (Virtanen et al., 2020). The center of the distribution can be selected individually as well as the peak width, determining the rate at which the probability changes. The probability $P(X)$ of a variable $X$ is given by:

$$P(X) = \frac{1}{2}\left(1 + \mathrm{erf}\left(\frac{X-\mu}{\sigma\sqrt{2}}\right)\right), \tag{2}$$

where $\mu$ represents the mean of the distribution and $\sigma$ the standard deviation. $\mu$ determines the center of the probability distribution. Values smaller than $\mu$ will have small probabilities, while values greater than $\mu$ will have large probabilities. $\sigma$ determines the width of the probability distribution. A larger value will result in a wider distribution, indicating a higher spread of the data. Conversely, a smaller value will lead to a narrower distribution. For the radar reflectivity factor, the center $\mu$ was set to -48 dBZ, and the width $\sigma$ to 6 dBZ. Furthermore, the probability distribution was inverted to ensure that low radar reflectivities lead to the highest haze echo probabilities. For the mean Doppler velocity, $\mu$ was set to -2 m s$^{-1}$, and the width $\sigma$ to 0.5 m s$^{-1}$.

 

Implementing a probability function for ceilometer measurements in Cloudnet is performed by using the following equation:

$$P(X) = \frac{\beta}{2\sigma\Gamma\left(\frac{1}{\beta}\right)} \exp\left[-\left(\frac{|X-\mu|}{\sigma}\right)^{\beta}\right], \tag{3}$$

where $\mu$ represents the mean of the distribution which in this case would mark the location of $100\,\%$ probability. The width of
the distribution is controlled by $\sigma$ and $\Gamma$ represents the gamma distribution. The shape of the distribution is defined by $\beta$. In
our measurements $\beta = 7$.

For the ceilometer measurements $\mu$ is set to $7.7 \times 10^{-6}\,\mathrm{m^{-1}\,sr^{-1}}$ and $\sigma$ is set to $2 \times 10^{-6}\,\mathrm{m^{-1}\,sr^{-1}}$. The choice of this
distribution ensures, that the probability for haze echo increases close to the value of $0.5 \times 10^{-6}\,\mathrm{m^{-1}\,sr^{-1}}$ and starts to decrease
for the threshold close to $0.5 \times 10^{-6}\,\mathrm{m^{-1}sr^{-1}}$ that is used to locate liquid cloud layers in Cloudnet. By performing element-wise
multiplication of the haze echo probability arrays for radar reflectivity factor, mean Doppler velocity and ceilometer attenuated
backscatter coefficient, the combined probability can be estimated. When the combined haze echo probabilities exceed $60\,\%$
for grid points below the CBH or at altitudes below $2\,\mathrm{km}$ (average top of the height of the marine aerosol layer over Barbados)
when no CBH is detected, the haze echo category is implemented and replaces targets previously classified as "Drizzle or rain"
in the Cloudnet target classification.

Figure 5 illustrates the haze echo probability estimation procedure for the same case study shown in Fig. 4. The detailed
discussion below shows that a haze echo probability based on a single variable is not suitable and that instead, the combined
haze echo probability leads to the most confident estimates of haze echo occurrence. In the radar and lidar observations at
the BCO shown here from 2 December 2021 between $06{:}00 - 08{:}00\,\mathrm{UTC}$, several cloud types were present: a precipitating
stratiform cloud between $06{:}15 - 07{:}00\,\mathrm{UTC}$ with CBH of around $2\,\mathrm{km}$, two deeper cumulus clouds with CTHs of around
$3.5\,\mathrm{km}$ and shallow cumulus clouds with CTHs below $1\,\mathrm{km}$. The black line in Fig. 5 denotes the CBH detected by Cloudnet.
Rain on the ground was observed from two short showers at around $06{:}25\,\mathrm{UTC}$ and $07{:}10\,\mathrm{UTC}$. The presence of virga is
visually evident through the distinctive fall streaks in the time series of the radar reflectivity factor in a) with weak radar
reflectivities at the edges of the fall streaks between $06{:}10 - 06{:}50\,\mathrm{UTC}$, a characteristic attributed to the evaporation of drizzle
drops. The estimated probability of sea salt aerosols (haze echoes) based on radar reflectivity alone in Fig. 5 (b) shows that
parts of the fallstreaks show probabilities over $60\,\%$.

High haze echo probabilities based solely on radar mean Doppler velocity observations in Fig. 5 (d) indicate upward motion
below the CBH between $06{:}00 - 06{:}10\,\mathrm{UTC}$ and after $07{:}10\,\mathrm{UTC}$ in Fig. 5 (c). The threshold for estimating haze echo proba-
bility based on radar mean Doppler velocity is chosen in a way to ensure that haze echoes, that are too small to overcome the
background motion, are also identified in downdrafts. It can be seen that when rain was observed at the ground, radar reflec-
tivities exceed approximately $0\,\mathrm{dBZ}$ at the lowest radar range gate between $06{:}20 - 06{:}30\,\mathrm{UTC}$ and the near-surface attenuated
backscatter coefficient (Fig. 5 (e)) is larger than the liquid CBH-threshold of $1.5 \times 10^{5}\,\mathrm{m^{-1}\,sr^{-1}}$ due to large drizzle drops that
dominate the signal. However, owing to the limited number concentration of drizzle- or raindrops relative to cloud droplets,
the ceilometer signal is not fully attenuated by the near-surface precipitation, so ceilometer-based CBH retrieval was - except
for times of stronger precipitation - still reliable. What is also evident in Fig. 5 (e) is that the ceilometer attenuated backscatter

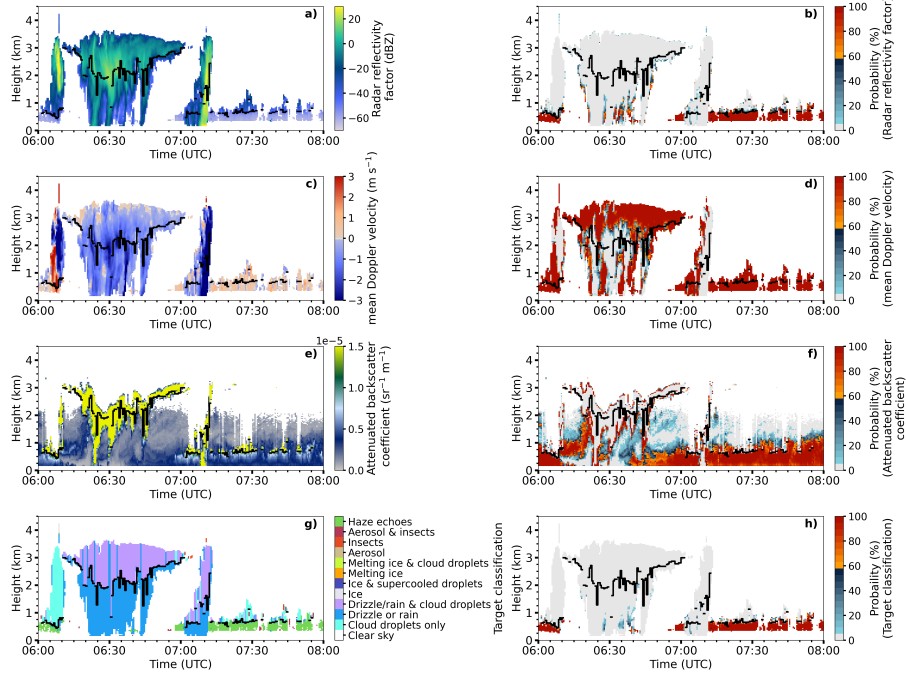

**Figure 5.** BCO case study 2 December 2021: Observations and corresponding sea salt probabilities for the radar reflectivity factor (a), (b), radar mean Doppler velocity in (c), (d), ceilometer attenuated backscatter coefficient (e), (f) and Cloudnet target classification (including the haze echo classification) (g)- and combined haze echo probability (h). Haze echoes are classified when the combined probability in (h) exceeds 60 %.

coefficient below the precipitating stratiform cloud is lower compared to the times when haze echoes occur below the shallow cumulus clouds. This can be attributed to the wet deposition of aerosol particles within the marine boundary layer, during or after precipitation periods that lead to comparatively cleaner atmospheric conditions. The resulting haze echo probability based on the thresholds of the attenuated backscatter coefficient can be seen in Fig. 5 (f). The choice of thresholds ensures that sea salt probabilities are low during and after times of precipitation as illustrated in the combined haze echo probability in Fig. 5 (h).

Haze echoes are classified for combined probabilities larger than 60 %. The resulting Cloudnet target classification with haze echoes as a new target class is shown in Fig. 5 (g). For the presented case study, the Cloudnet classification of "Drizzle or rain" is now limited to times when the precipitation-producing stratiform cloud and the deeper trade wind cumuli (around 07:00 – 07:10 UTC) occur and misclassifications of "Drizzle or rain" below the shallow trade-wind cumuli are now replaced by the new class "haze echo".

## 3.2  Cloud type classification

The number of detected cloud profiles per time step for profile-based methods depends on the number of detected CBH. If information about the CBH is missing, although the radar detects signals from clouds, cloud profiles for profile-based methods





such as the Virga-Sniffer tool or in Cloudnet are not registered. Moreover, to distinguish between clouds by altitude of their CBH, profile-based approaches could classify individual cloud profiles within a single cloud object as different cloud types if the CBH varies strongly within the cloud.

Continuous radar signals in time and space are detected as a cloud object in object-based detection methods. Once a cloud object has been detected, information about the CBH can be analyzed within this object. In the case that gaps occur in the CBH measurements of the ceilometer (e.g. during heavy precipitation), still a single cloud is detected and not multiple clouds. In addition, more cloud profiles are recognized in multi-layer cloud situations if the ceilometer signal is attenuated. As a result, a higher number of cloud profiles are automatically detected by object-based detection methods. A disadvantage of object-based cloud detection methods is that multiple contiguous cloud objects (when clouds touch at the cloud edges or grid points overlap) are recognized as a single cloud object. This can be the case, for example, when fall streaks of a stratiform cloud fall into an underlying trade wind cloud. In Schulz et al. (2021), for example, a combined cloud category was introduced for such situations as "a mixture of cumulus with an attached stratiform layer" (StSc + Cu) or in Lamer et al. (2015) the category "precipitating cumulus with stratiform outflow" (Precip. Cu.Str.).

To investigate precipitation properties of and haze echo occurrence below warm clouds and trade wind cumuli (as a subset of warm clouds) separately, an object-based cloud classifier was developed. Specifically, after assigning haze echoes using the updated Cloudnet target classification with the method explained in Section 3.1, a cloud type classification was performed using object-based feature detection methods from the SciPy library (Van der Walt et al., 2014).

A cloud is detected for connected radar signals (hydrometeor clusters) with a minimum size of at least three connected pixels. Similar to similar to Kalesse-Los et al. (2023), we group individual clouds by their CBH into three cloud categories (Fig. 6), namely warm clouds, warm clouds with CBH below 1 km (i.e., trade wind cumuli), and cold clouds. Warm clouds are defined as clouds with bases and tops below the height of the 0 °C wet-bulb temperature isotherm that is taken from data of the Cloudnet target categorization. Please note that Kalesse-Los et al. (2023) simply relied on CBH lower than 4 km for warm-cloud identification (according to mean freezing levels of 4.8 km over RV Meteor during EUREC[4]A) which will lead to differences in statistical results compared to our study. Trade wind clouds are defined as clouds with CBH below 1 km and a cloud top below the height of the 0 °C wet-bulb temperature isotherm. Cold or ice-containing clouds are usually defined as clouds with a cloud top above the height of the 0 °C wet-bulb temperature isotherm. The temperature criterium is based on the Cloudnet target categorization scheme as falling ice melts when the wet-bulb temperature becomes positive rather than the actual air temperature (Hogan and O'Connor, 2004). Lidar-derived ice-nucleating particle concentration over Barbados indicates that immersion freezing starts to take place at temperatures lower than -10 °C (Haarig et al., 2019; Harrison et al., 2022). BCO Cloudnet observations during EUREC[4]A show that deeper trade wind cumuli often had tops just barely exceeding the height of the 0 °C-wet-bulb temperature isotherm. Thus, for the comparison of the EUREC[4]A datasets obtained at BCO and RV Meteor, the temperature criterium was relaxed to -10 °C to enable a higher comparability to the cloud statistics over RV Meteor derived by Kalesse-Los et al. (2023).

The cloud classification depends on the accurate estimation of the CBH and CTH. Cloud top height is determined by the highest located radar pixel of a hydrometeor cluster that can be estimated from the location of the bounding box (Fig. 6) of the



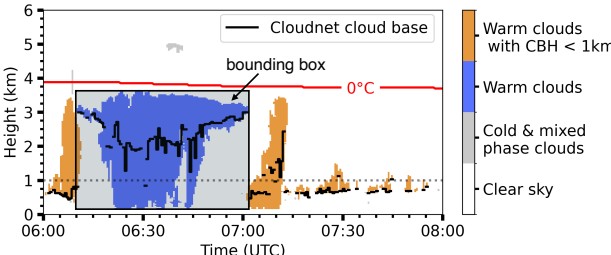

**Figure 6.** BCO case study 2 December 2021: Example of the cloud type classification. Clouds are characterized as "cold & mixed-phase clouds" when their cloud top height (CTH, height of the top of the bounding box) is located above the altitude of the $0\,°C$ of the wet-bulb temperature isotherm (red line). "Warm clouds" have a cloud base height (CBH) and CTH below the altitude of the $0\,°C$ wet-bulb temperature isotherm and are classified as "trade wind cumuli" when their CBH is located below $1\,km$. The cloud base height (from Cloudnet) is denoted by the black line when detected.

cluster. The first hydrometeor base height is defined by the lowest-located radar pixel of a hydrometeor cluster (taken from the horizontal lower bound of the bounding box). The radar signals of the first hydrometeor base height can either originate from a cloud or from precipitating hydrometeors. Warm clouds objects that are precipitating with a majority of detected cloud bases 335 below $1\,km$ are defined as trade wind clouds. Cloud objects with a majority of detected cloud bases above $1\,km$ are assigned as warm clouds. This ensures, that precipitating stratiform clouds or clouds accompanied by outflow layers near cumulus tops are not defined as trade wind clouds due to the height of the first hydrometeor base in case they are precipitating. An example of a precipitating stratiform cloud with a hydrometeor base below $1\,km$ and the majority of detected CBH above $1\,km$ is illustrated 340 in Fig. 6 between $06:10 - 07:00$ UTC.

## 4 Results

Impacts of environmental factors on seasonal haze echo occurrence are presented in Sect. 4.1. Section 4.2 is comprised of a statistical analysis of the extended Cloudnet target classification including "haze echoes" with a focus on clouds below the freezing level for two full years between 1 July 2021 and 1 July 2023 (508 days of data) and during the EUREC[4]A campaign 345 over the BCO (18 January 2020 to 19 February 2020, 28 days of data). Virga-Sniffer statistics are shown in Sect. 4.3 to validate the occurrence of haze echoes in the measurements over the BCO as identified by the proposed combined haze echo probability method incorporated into the Cloudnet target classification. Finally, limitations of the proposed haze echo classification for sea salt aerosol occurrence are discussed in Sect. 4.4.

### 4.1 Impact of environmental factors on seasonal changes in haze echo occurrence

Following the method proposed by Klingebiel et al. (2019), a dimensionless haze echo occurrence parameter is used, representing the cumulative sum of all pixel per time step where haze echoes are detected. To test the influence of environmental factors, haze echo occurrence is set into context with $2\,m$ surface observations of wind speed, wind direction, and relative humidity.



Both, the haze echo occurrence data and the 2 m-surface observations are averaged over a 6 h time window to be in line with previous studies. This time window balances maintaining consistency in observed air masses with ensuring a sufficiently large

statistical sample (Nuijens et al., 2009; Klingebiel et al., 2019).

Analyses are performed separately for the dry and wet season over Barbados. The seasonal changes in cloud cover over Barbados can be attributed to the migration of the Intertropical Convergence Zone (ITCZ). During the dry season (December to June), the ITCZ is furthest away from the island and the region experiences strong subsidence. During the wet season (June to December), the low-altitude convergence favors deep convection. Seasonal differences in cloud cover and the occurrence of

360 precipitation events are therefore mainly characterized by the higher proportion of deep convective events with high rainfall rates during the wet season (Stevens et al., 2016).

Figure 7 shows the haze echo occurrence for the dry and the wet season for two years (July 2021 until July 2022 and July 2022 until July 2023). Occurrences of haze echo larger than two (representing deeper haze echo layers) are more frequent during the dry season compared to the wet season (Fig. 7 (a,b)). Interestingly, a haze echo occurrence larger than two is less frequent

in 2022/23 for both seasons compared to 2021/22. This can be attributed to lower wind speeds in 2022/23 (Fig. 7 (c,d)) since wind speed is known to have the strongest effect on the production of sea spray (Lewis and Schwartz, 2004). Furthermore, the variation in wind direction is higher in 2022/23 compared to 2021/22, especially for the dry season. Remarkably, wind direction during the dry season of 2022/23 (Fig. 7 (e)) indicates that the trade winds were also less constant and showed a similar variation typical for the conditions during the wet season. What also stands out, is that surface relative humidity for both, the dry and

wet seasons is much higher in 2022/23 compared to 2021/22 (Fig. 7 (g,h)). Higher surface relative humidity results in lower cumulus cloud base height leading to lower haze echo occurrences in the sub-cloud profiles. Higher near-surface relative humidity, lower wind speeds, and more southerly wind directions over Barbados are associated with the northward shift of the ITCZ and a weakening of the subtropical high in the northeastern North Atlantic. Therefore, the meteorological conditions in the first half of 2023 at BCO are likely related to a remote El Niño-Southern Oscillation response (Brueck et al., 2015). It is

important to acknowledge that this analysis focused on specific variables, and there might be other unaccounted factors that can influence the occurrence of haze echoes. More comprehensive investigations are needed to fully understand the factors influencing haze echo occurrence and their seasonality.







**Figure 7.** 6 h-mean haze echo occurrence (bars) in (a) and (b) and 6 h-mean 2 m-surface observation of wind speed in (c) and (d), wind direction in (e) and (f) and relative humidity in (g) and (h) for the dry and the wet season. The two-year data set was divided into two periods spanning from July 2021 until July 2022 (labeled as 2021/2022) and July 2022 until July 2023 (labeled as 2022/23).





## 4.2 Cloudnet statistics including haze echoes

Seasonal changes in cloudiness, precipitation, and haze echo occurrence for warm clouds and the warm cloud subset trade wind
cumuli are evident in the measurements at the BCO between the dry and wet seasons from 1 July 2021 to 1 July 2023 in Fig. 8
and can be analyzed by using the proposed enhanced Cloudnet target classification. Here, "Liquid droplets" (in Fig. 8 (a)) refer
to the combined Cloudnet target classifications "Cloud droplets only" and "Drizzle/rain & cloud droplets".

In the dry season, the frequency of occurrence of liquid droplets is on average higher (for warm clouds and trade wind
cumuli) compared to the wet season. The relative frequency of "Liquid droplets" in warm clouds and trade wind cumuli peaks
near the LCL at 793 m in the dry season and 669 m in the wet season. Due to higher surface relative humidities during the wet
season, the average height of the LCL is lower which shifts the CBH downwards (Nuijens et al., 2015). The heights of these
maxima are within the range of the average heights of the cloud bases during the dry and wet season derived by Nuijens et al.
(2014). In general, the shape of the distribution of "Liquid droplets" from trade wind cumuli reflects the presence of shallow
cumulus humilis with cloud tops near $1.0 - 1.5$ km as well as deeper cumuli with cloud tops up to $2 - 4$ km as discussed by
Nuijens et al. (2014). The secondary maximum of "Liquid droplets" in warm clouds at altitudes between $1.5$ and $2.0$ km
indicates the presence of stratiform clouds below the trade inversion. Thus, the seasonally higher occurrence of warm clouds
in the dry season compared to the wet season is caused by a higher proportion of trade wind cumuli and of stratiform clouds
with CBH above $1$ km.

"Drizzle or rain" is more frequent during the dry season compared to the wet season (Fig. 8 (b)), which can be attributed to
the higher frequency of occurrence of warm clouds during the dry season. From the location of the peak in the "Drizzle or
rain" relative frequency at $700$ m during the dry season, the relative frequency of "Drizzle or rain" decreases by around $7\%$
(from $10\%$ to $3\%$) towards the height of the first radar range gate, indicating that a large proportion of "Drizzle or rain" from
warm clouds evaporates before reaching the ground. A similar shape in the vertical relative frequency distribution of "Drizzle
or rain" is evident for trade wind cumuli. In the wet season, subcloud evaporation of "Drizzle or rain" is still evident, albeit less
pronounced.

Haze echo distributions in the presence of warm clouds and trade wind cumuli are identical (Fig. 8 (c)). Haze echoes occur
a lot less frequently during the wet season compared to the dry season which can be attributed to the prevalence of different
environmental conditions as detailed in Sect. 4.1. Moreover, the interplay between dust transport and precipitation leads to the
wet scavenging of aerosols and very clean periods despite the higher dust loading during the wet season (Stevens et al., 2016).
Specifically, the haze echo relative frequency was found to decrease by a factor of four from about $16\%$ in the dry season to
$4\%$ in the wet season. At the same time, haze echo layers in the wet season are on average shallower than during the dry season
which is substantiated by Fig. 7 (a,b). These results also answer the question how frequent "Drizzle or rain" is misclassified in
the standard Cloudnet target classification for the BCO: Fig. 8 (c) shows that the frequency of occurrence of "Drizzle or rain"
in Cloudnet for the BCO without filtering haze echoes as proposed here is overestimated by up to $16\%$ (at $482$ m altitude) in
the dry season. Please note that an analysis of the haze echo occurrence during EUREC[4]A was also performed. As the results
are very similar to the presented long-term statistics during the dry season, they are not displayed in Fig. 8.





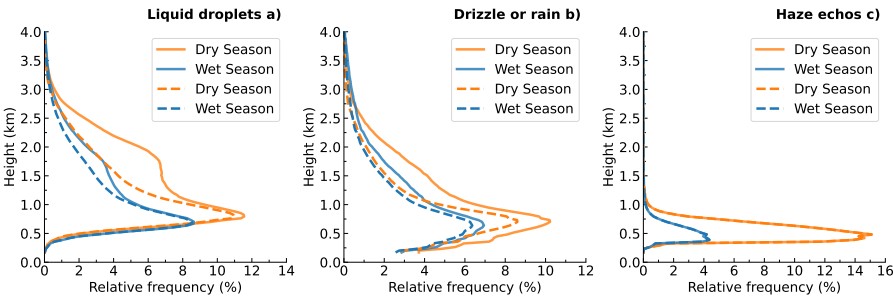

**Figure 8.** Statistics of enhanced Cloudnet target classification statistics including the "haze echo" target class for BCO based on data from 1 July 2021 until 1 July 2023: Liquid droplets (a), Drizzle or rain (b) and haze echoes (c). Results for warm clouds (solid line) and trade wind cumuli (dashed line) for the dry and wet season are shown.

## 4.3 Virga-Sniffer results including haze echoes

The Virga-Sniffer enables the detection of precipitation and haze echoes and is used here as an independent tool that validates our haze echo detection method. To ensure that the Virga-Sniffer for the BCO is configured optimally, we establish a statistical

comparison of the results of the Virga-Sniffer at the BCO, with the results of Kalesse-Los et al. (2023) during the EUREC[4]A campaign. Please again note the following differences between the study by Kalesse-Los et al. (2023) and this present study: In Kalesse-Los et al. (2023), warm clouds are defined as clouds with a CBH below 4 km and trade wind cumuli are defined as clouds with a CBH below 1 km. Furthermore, a profile-based approach (as in Cloudnet) is used. For the BCO measurements however, we differentiate between warm clouds and trade wind cumuli using an object-based detection method described in

Sect. 3.2. In addition, we define warm clouds as clouds with CTH is below the height of the 0 °C isotherm. To relate the following statistics and allow a better comparison between the BCO and RV *Meteor* observations, the object-based detection method was also applied to the RV *Meteor* observations during the EUREC[4]A campaign. Also, for the first time, we obtain long-term statistics from the Virga-Sniffer at the BCO.

### 4.3.1 EUREC[4]A statistics at the BCO and the RV *Meteor*

Figure 9 (and Table D1 in Appendix D) gives an overview of Virga-Sniffer derived statistics of clouds, precipitation, and haze echoes for BCO and RV *Meteor* during EUREC[4]A. The RV *Meteor* profile-based results show an 8 % lower proportion of warm clouds compared to the RV *Meteor* object-based results. This difference can be explained by two facts: One, in the object-based approach, the temperature threshold to differentiate between warm and cold clouds is -10 °C. Using a temperature criterium of 0 °C, reduces the proportion of warm clouds to 76 % and the proportion of rain reaching the ground to 7 % in this case. Second,

within the profile-based approach, only single profiles with CBH below 4 km are considered as warm clouds. In Kalesse-Los et al. (2023) the Virga-Sniffer tool is configured to replace the lowest CBH layer by the calculated LCL. Consequently, also profiles with gaps in the CBH layer (e.g. due to strong attenuation of the ceilometer by rain) are considered in the profile-based statistic. This would also explain the higher proportion of trade wind cumuli in Kalesse-Los et al. (2023) compared to the





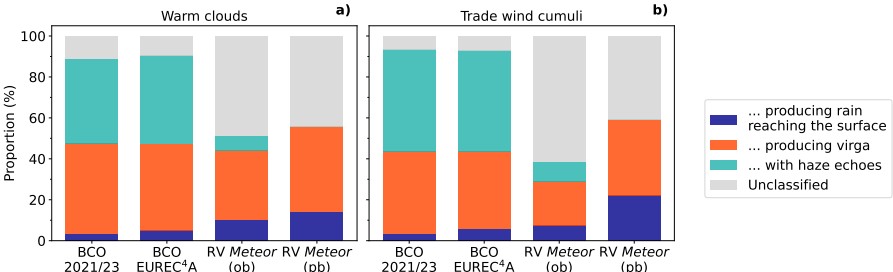

**Figure 9.** Virga-Sniffer based statistics of precipitation, virga and haze echo in % for warm clouds and trade wind cumuli for different observation periods and platforms. For the BCO, the object-based (ob) cloud classification was applied for the long-term measurements and EUREC[4]A. The (ob) results over RV *Meteor* during EUREC[4]A are contrasted to the profile-based (pb) statistics taken from Kalesse-Los et al. (2023). The category "Unclassified" in this context includes all other radar signals below the CBH as well as profiles with clouds that contain neither precipitation nor haze echoes below the cloud base.

object-based results for the RV *Meteor*. The comparison of profile- and object-based approaches illustrates the influence of the Virga-Sniffer configuration on the statistical results. The influence of the Virga-Sniffer configuration is analyzed in more detail in Appendix A.

Comparing BCO and RV *Meteor* object-based statistics, the proportion of precipitation reaching the ground is higher over the RV *Meteor* (5 % vs. 10 %, cf. Fig. 9 (a) and Table D1). Comparability of these statistics is limited though because of the following: Precipitation reaching the surface is classified when a reflectivity threshold of 0 dBZ is exceeded in the lowest radar range gate. The lowest radar range above RV *Meteor* is about 150 m higher than that over the BCO. Thus, over BCO the drizzle drops can fall a longer distance in the dry sub-cloud layer before reaching the lowest radar range gate and could thus evaporate to smaller sizes so that the surface rain reflectivity threshold is reached less frequently at the BCO.

Above the BCO, the proportion of virga is higher compared to the RV *Meteor*. The higher proportion of virga could again be explained by the reflectivity threshold for surface precipitation but also by the higher sensitivity of the radar at the BCO. This mainly involves radar reflectivities below -50 dBZ, which can be caused by sea salt aerosols or evaporating drizzle drops. Therefore, the virga proportion at BCO may also be overestimated.

### 4.3.2 Longterm statistics at the BCO

To assess the credibility of the developed haze echo detection method, we conducted a comparison with the Virga-Sniffer haze echo detection for measurements between July 2021 to July 2023 over the BCO. For all pixels classified as haze echoes by the newly proposed haze echo identification method, the Virga-Sniffer classified a proportion of 72 % as haze echoes, 7 % as clouds, 19 % as virga, and 0 % as precipitation reaching the ground. These results indicate that both methods compare reasonably well. However, better agreement of haze echo detections can be achieved when modifying the Virga-Sniffer configuration as explained in Sect. 4.4.



Between July 2021 and July 2023 over the BCO, 61 % of all detected clouds are warm clouds (see Fig. 9 and Table D1 in
Appendix D). 48 % of these warm clouds produce precipitation, from which 44 % fully evaporates before reaching the ground.
68 % of warm cloud virga originate from trade wind cumuli. When not counting haze echoes while virga is detected in the
same profile, the Virga-Sniffer identifies haze echoes below the CBH of warm clouds during 41 % of the time. When we
include non-precipitating and precipitating warm clouds (i.e. also include profiles where haze echoes and virga are detected
simultaneously), haze echoes occur 64 % of the time. In a two-year study over the BCO by Klingebiel et al. (2019), haze
echoes were found to occur in 76 % of the time steps and are thus somewhat larger compared to the results obtained with the
Virga-Sniffer. Please note two main differences in these statistics though: Klingebiel et al. (2019) only counted haze echoes as
being valid when the number of haze echo pixels in a profile was greater than four. In contrast, in our study, we count haze
echoes even if they occur only once per profile Furthermore, haze echoes in Klingebiel et al. (2019) are also counted when no
CBH is detected. These time steps are not considered in our statistics.

For the considered two-year time period at BCO, 75 % of the detected warm clouds are trade wind cumuli. This result is
higher than that of Nuijens et al. (2014), who estimated that the contribution of cloud cover near the LCL is about two-thirds of
the total cloud cover of clouds below 4 km. The temporal cloud cover in their study refers to the proportion of time that a CBH
is detected above the observation site. Consequently, cloud cover near the LCL should be similar compared to the proportion
of trade wind cumuli but not compared to the warm cloud proportion as the contribution of stratiform clouds to the cloud cover
is not fully reflected in the results from Nuijens et al. (2014). Furthermore, in Nuijens et al. (2014) an approx. 10 dBZ higher
reflectivity threshold (of -40 dBZ) was applied to the radar reflectivity factor for cloud identification. They found that lowering
the reflectivity threshold increases the number of detected optically thin clouds near the LCL. Furthermore, the Virga-Sniffer
was configured to perform best for warm cloud identification. Accordingly, the proportion of warm clouds is overestimated
here. A discussion on this can be found in Appendix B. Considering these differences in approaches, our results are in a similar
range as the ones of Nuijens et al. (2014).

Intra-year variability at BCO for the two analyzed years is high as illustrated in Fig. 10. Specifically, the proportion of rain
reaching the surface is reduced in the dry season in 2022/23 compared to the dry season in 2021/22, while the decrease in the
proportion of virga is less pronounced. The lower proportion of rain reaching the surface could be related to a lower proportion
of deep trade wind cumuli. When trade wind cumuli are deep (reach heights between 2 – 4 km), they rain more frequently
(Nuijens et al., 2014). As surface observations indicate, relative humidities were in general higher, and wind speeds lower in
2023 (see Fig. 7). Lower wind speeds could be an explanation for why CTHs of trade wind cumuli reached lower heights in the
dry season of 2022/23, according to (Brueck et al., 2015). The lower proportion of raining trade wind cumuli may therefore be
related to the meteorological conditions in 2022/23.

The proportion of warm clouds that are trade wind cumuli is notably higher during the wet season, indicating a lower amount
of stratiform outflow cloud layers during this season. These findings support the ones of Nuijens et al. (2014) who found that
cloud cover above 1 km is larger during the dry season due to the higher occurrence of deeper clouds with tops near the
inversion and stratiform outflow which is substantiated by Fig. 8 and Fig. 10 (a). Moreover, their results indicate that fewer



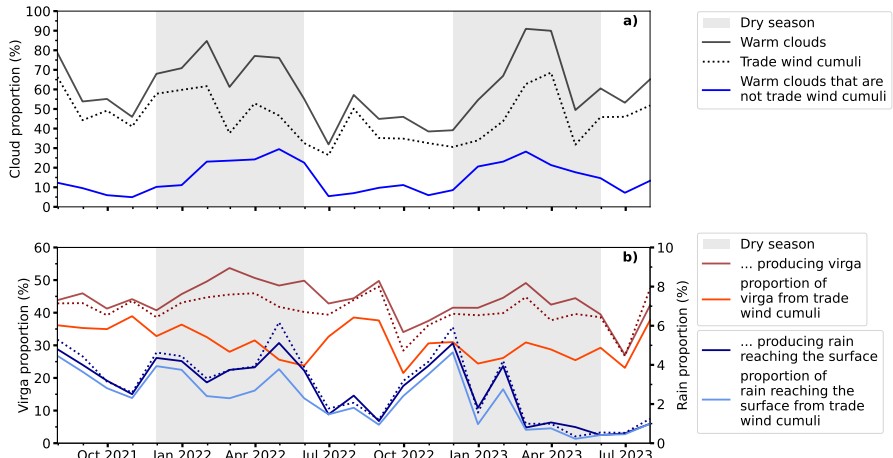

**Figure 10.** Monthly mean proportion of warm clouds, trade wind cumuli and warm clouds that are not trade wind cumuli in panel (a) and virga as well as rain reaching the ground for warm clouds and trade wind cumuli (dotted line) in panel (b) over the BCO from July 2021 to July 2023. The dry season is indicated by the grey-shaded area that spans from December until June, respectively.

trade wind cumuli are reaching heights of $2-4\,\mathrm{km}$ during the wet season. A higher proportion of stratiform outflow clouds in the dry season might be the reason for the higher proportion of virga from warm clouds in that season.

## 4.4 Limitations of the proposed haze echo identification method

As shown by Klingebiel et al. (2019), it is most likely that haze echoes are caused by hygroscopically grown sea salt particles and not by precipitation. However, we want to discuss the possibility that signals we classify as haze echoes are caused by drizzle drops that fall into strong updrafts, evaporate, and decrease in size indicated by a downward decrease of radar reflectivity factor.

The radar reflectivity factor is proportional to the sixth power of the drop diameter. A few large drops can therefore lead to large radar reflectivity values. In that case, high radar reflectivity factors within the cloud would indicate the presence of drizzle drops. Typical threshold values for the presence of drizzle in a cloud range between -20 and -15 dBZ (Kogan et al., 2012; Frisch et al., 1995). For a case study at the BCO, Acquistapace et al. (2019) showed that embryonic drizzle drops form when radar reflectivity factors around -30 dBZ are detected within a cloud. In their study, skewness was used as a criterion for the detection of drizzle formation. The value of -30 dBZ indicates the potential presence of embryonic drizzle drops, while drizzle drops that are large enough and dominate the signal in the radar measurements have radar reflectivity factors of around -10 dBZ (Acquistapace et al., 2019). The BCO data available to us is missing information about the radar Doppler spectrum, which is why higher moments such as skewness could not be determined. We investigate the radar reflectivity values for shallow cumulus clouds with CBH below $1\,\mathrm{km}$ and CTH below $2\,\mathrm{km}$ between July 2021 and July 2022. Haze echoes occur in $52\,\%$ of all profiles below the CBH of shallow trade wind cumulus. Our analysis shows that all radar reflectivity values within these



clouds are lower than -10 dBZ and thus do not indicate the presence of drizzle. Only 17 % of all radar reflectivity factors in shallow cumuli range between -30 dBZ and -10 dBZ. Accordingly, a clear majority (83 %) of all pixels within shallow cumulus clouds in our study have radar reflectivity values that do not indicate the presence of drizzle formation.

A study by Albright et al. (2023) at the BCO further supports the idea that very shallow clouds over Barbados rarely produce precipitation. From their observations, they hypothesize that a large part of the condensate from clouds, that form within the transition layer between 550 and 700 m and have CTH below 1.3 km, evaporates as the role of these clouds is to humidify the transition layer. In the two-year dataset analyzed here, we also detect haze echoes below these very shallow clouds and in-line with Albright et al. (2023) conclude that for these clouds, misclassification of precipitation as sea salt aerosols is unlikely.

For deeper cumulus clouds (i.e. warm clouds) with a cloud base of below 1 km, haze echoes determined with the method proposed here occur in 36 % of all profiles below the CBH. 27 % of all pixels of deeper cumulus clouds with haze echoes below their CBH show radar reflectivity values greater than -30 dBZ. For all deeper cumulus clouds where no haze echoes are detected below CBH, 59 % have radar reflectivity factors that are greater than -30 dBZ. The tops of these deeper cumulus clouds reach greater altitudes compared to those with haze echoes below CBH. Consequently, these clouds are more likely to precipitate. From these results, we conclude that in deeper cumulus clouds with haze echoes below the CBH, the possibility

exists that in a few cases, precipitation and haze echoes can occur together in one profile. Consequently, only for a minority of cases, our method might not be able to discriminate between drizzle and haze echoes.

    The optimal configuration of the Virga-Sniffer is a prerequisite for comparing its results with those of the developed method for the detection of sea salt aerosols. The proportion of time steps classified as virga for pixels classified as haze echoes by our method is influenced by the configuration of the clutter mask and velocity mask of the Virga-Sniffer. The clutter mask

determines the number of potential grid points that can be identified as virga for negative Doppler velocities. The choice of the velocity mask and thresholds of the Virga-Sniffer configurations restricts the occurrence of virga to negative Doppler velocities. Consequently, precipitation is not identified in updrafts indicated by positive Doppler velocities. Changing the clutter mask to $m = -55$ and $c = -38$ reduces the proportion of identified virga by the Virga-Sniffer for classified haze echo pixels to 0 %. The proportion of haze echoes identified with the developed method and by the Virga-Sniffer would in this case increase to

89 %. This largely explains the differences in the proportion of detected haze echoes between the method developed here and that of the Virga-Sniffer. The differences demonstrate the uncertainties of both methods, which cannot reliably distinguish between drizzle and haze echoes when they potentially co-occur.

## 5   Conclusions

Here, we build upon the study by Klingebiel et al. (2019), which concluded that haze echoes over the BCO are caused by

535 hygroscopically grown sea salt particles. A method has been developed to discriminate between haze echoes and "Drizzle or rain" in Cloudnet, utilizing a combined probability threshold based on cloud radar reflectivity factor, radar mean Doppler velocity and ceilometer attenuated backscatter coefficient. Our method, which works similar to the insect detection algorithm in Cloudnet, relies on the appropriate selection of thresholds as inputs for deriving heuristic probability functions. We do



acknowledge that the method has limitations, and some residual misclassification may remain in Cloudnet, particularly in cases where small drizzle drops fall into a layer of haze echoes. We use an object-based detection method that distinguishes between warm clouds (CBH below 4 km and CTH below the 0 °C-wet bulb temperature isotherm) and trade wind cumuli (CBH below 1 km).

The results of the Cloudnet statistics with the method developed here for a two-year period from July 2021 to July 2023 focus on warm clouds over the BCO which were selected from the observations using the described object-based approach. The vertical distribution of "Liquid droplets" (being the combination of the target classes "Cloud droplets only" and "Drizzle or rain & cloud droplets") peaks at specific altitudes corresponding to CBH of shallow cumulus clouds during both seasons. Trade wind cumuli are more frequent in the dry season, and warm clouds show a second increase in relative frequency due to the presence of stratiform outflow clouds below the trade inversion between 1 and 3 km. A higher occurrence of "Drizzle or rain" in the dry season can therefore be explained by the higher occurrence of warm clouds. In the dry season, undetected haze echoes in the standard Cloudnet target classification, were found to lead to an overestimation of up to 16 % of "Drizzle or rain" occurrence. Significant inter-annual differences as well as seasonal differences in haze echo occurrence were found. Seasonal differences in the occurrence of haze echoes over the BCO, are related to changes in wind speed and wind direction. Haze echoes show an increased occurrence in the dry season when wind speeds are higher compared to the wet season. This supports existing assumptions that indicate a connection between sea salt aerosols and wind speed and supports the hypothesis that haze echoes originate from sea salt aerosols.

To validate our method, we applied the Virga-Sniffer tool to the measurements over the BCO. 72 % of all pixels identified as haze echoes by our method are also identified as haze echoes by the Virga-Sniffer tool, which demonstrates that haze echoes are detected by two independent methods. Modifications of virga sniffer settings resulted in even better comparabilities of haze echo occurrences. In addition to the Cloudnet long-term statistics at the BCO, we compare the Virga-Sniffer statistics with those of Kalesse-Los et al. (2023) during the EUREC[4]A campaign.

Between July 2021 and July 2023, 61 % of all detected clouds over the BCO are warm clouds, with 48 % producing precipitation. Most of the precipitation (44 %) was found to fully evaporate before reaching the ground. Additionally, 45 % of all clouds are trade wind cumuli, with 44 % of them precipitating, and 40 % of the precipitation fully evaporating before reaching the surface. Comparisons between the dry and wet seasons reveal higher proportions of warm clouds during the dry season, while trade wind cumuli proportions remain similar. During EUREC[4]A, warm clouds occur in 71 % of the time, with proportions of virga similar to long-term statistics. For warm clouds, the proportion of virga is similar over the BCO compared to the RV *Meteor* (47 % compared to 44 %). A lower proportion of precipitation reaching the ground from warm clouds is observed at the BCO compared to the RV *Meteor*.

The presented method is applicable to all marine atmospheric remote-sensing facilities that provide a 35 GHz cloud radar, and a ceilometer. The method to identify haze echoes in the Cloudnet target classification depends on the selection of thresholds. Other atmospheric observatories with similar instrumentation close to the ocean that might detect sea salt aerosols need to select appropriate thresholds dependent on the characteristics of their instruments and individual calibration factors. All in all, our proposed method for haze echo identification significantly improves the classification of "Drizzle or rain" in the Cloudnet



target classification scheme in the measurements over the BCO and was independently validated with the Virga-Sniffer tool. For the first time, long-term statistics from the Cloudnet dataset and the Virga-Sniffer were carried out at the BCO. The conducted statistics therefore form a basis for future studies that quantify the occurrence of precipitation and the subcloud fate of precipitation in the trade wind regions. Haze echoes as identified here serve as a proxy for sea salt aerosol occurrence which opens the possibility of further in-depth sea salt-cloud interaction studies.

## Appendix A: Virga-Sniffer configurations

The Virga-Sniffer is freely configurable by selecting different flags and threshold values to recognize virga from the given input data. The default configuration in Table A1 (configuration 0) summarizes all configuration flags, thresholds, and settings used for the EUREC$^4$A dataset from the RV *Meteor* described in (Kalesse-Los et al., 2023). Configuration 1 contains the selection of the flags and threshold values selected for the BCO data set. The cloud and precipitation statistics between the object-based and profile-based methods show considerable differences. To increase the comparability of the two methods, two additional configurations were tested for the EUREA$^4$A data set from RV *Meteor* (configurations 2 and 3). The impact of the two configurations on the Virga-Sniffer statistics is discussed in Appendix C. A complete description of the individual configuration parameters can be found in the Virga-Sniffer documentation (Witthuhn et al., 2022).

## Appendix B: Cloud base height detection by the Virga-Sniffer

The configurations of the Virga-Sniffer tool are configured to detect warm clouds. A comparison between the CBHs detected with the Virga-Sniffer tool and the number of time steps in which a cloud object was detected can be seen in Table B1. From July 2021 to July 2023, the Virga-Sniffer identified 93 % of all CBHs from trade wind cumuli and 85 % of warm clouds. 40 % of all clouds with bases or tops above the height of the 0 ° wet-bulb temperature isotherm (cold clouds) are recognized by the Virga-Sniffer. There are likely two reasons for the lower proportion of detected cold clouds. Firstly, the ceilometer signal is attenuated in liquid layers. Consequently, in multi-layered cloud situations, the ceilometer detects the cloud base of trade wind cumuli and misses a larger proportion of CBHs from stratiform clouds, mixed-phase clouds, or ice clouds that lie above in the same profile. With the Virga-Sniffer tool, it is possible to fill gabs in cloud bases as part of the CBH preprocessing (see Table A1). The `cbh_fill_limit` is set to 60 s in this study, whereby CBH gaps smaller than this time window are filled by linear interpolation (`cbh_fill_method`). Increasing the time window thus increases the detection range, but can also lead to unphysical CBH results and false-positive detection of virga (Kalesse-Los et al., 2023). The second explanation for the lower proportion of cold clouds detected by the Virga-Sniffer tool is therefore related to the configuration chosen for the BCO dataset.



**Table A1.** Virga-Sniffer configurations: configuration 0 refers to the original configuration for the RV *Meteor* used in Kalesse-Los et al. (2023) and configuration 1 was taken for the processing of the Virga-Sniffer data for the BCO dataset. Configurations 2 and 3 are optional and will lead to more comparable results when using the profile-based compared to the object-based statistical approach developed in this study.

| configuration | | 0 | 1 | 2 | 3 |
|---|---|---|---|---|---|
| **Flags** | | | | | |
| cbh_connect2top | | False | False | False | False |
| lcl_replace_cbh | | True | False | False | False |
| require_cbh | | True | True | True | True |
| mask_clutter | | True | True | True | True |
| mask_rain | | True | True | True | True |
| mask_rain_ze | | True | True | True | True |
| mask_vel | | True | True | True | True |
| **Cloud base preprocessing** | units | threshold values | | | |
| cbh_smooth_window | s | 60 | 60 | 15 | 15 |
| lcl_smooth_window | s | 300 | 300 | 300 | 300 |
| cbh_layer_thres | m | 500 | 1000 | 500 | 500 |
| cbh_clean_thres | % | 0.05 | 0.02 | 0.05 | 0.01 |
| cbh_fill_limit | s | 60 | 60 | 60 | 60 |
| minimum_rangegate_number | - | 2 | 2 | 2 | 2 |
| **Special configurations** | | | | | |
| cbh_fill_method | | slinear | slinear | slinear | slinear |
| cbh_processing | | [2, 1, 3, 1, 4, 2, 1, 3, 1, 4, 5] | [ 2, 0, 1, 0, 2, 0, 3, 1, 0, 2, 0, 3, 4 ] | [2, 1, 3, 1, 4, 2, 1, 3, 1, 4, 5] | [1, 0, 2, 0, 1, 0, 2, 4] |
| **Virga-detection-specific** | units | threshold values | | | |
| cloud_max_gap | m | 150 | 150 | 150 | 150 |
| precip_mask_gap | m | 700 | 200 | 700 | 700 |
| vel_thres | ms$^{-1}$ | 0 | 0 | 0 | 0 |
| ze_thres | dBZ | 0 | 0 | 0 | 0 |
| clutter_c | ms$^{-1}$ | -8 | -6 | -8 | -8 |
| clutter_m | ms$^{-1}$ | 4 | -8 | 4 | 4 |

**Appendix C:  Comparison between object-based and profile-based Virga-Sniffer statistics**

A comparison between the Virga-Sniffer statistics on the RV *Meteor* was performed to evaluate the detection of warm clouds and trade wind cumuli using both object-based (ob) and profile-based (pb) methods. This analysis aimed to determine differ-





**Table B1.** Proportion of different cloud classes in % from the Virga-Sniffer compared to the number of time steps where a cloud object was detected. Cold clouds denote clouds with CBH or CTH above the height of the 0 °C wet-bulb temperature isotherm. Periods of observation from July 2021 until July 2022 (2021/22) with the wet and dry season for this period are below. For July 2022 until July 2023 (2022/23) the same applies for the wet and the dry season. The two years from July 2021 until July 2023 are denoted as "2021/23".

|  | Trade wind cumuli | Warm clouds | Cold clouds | All clouds |
|---|---|---|---|---|
| **2021/22** | **96** | **86** | **42** | **63** |
| Wet season | 100 | 89 | 42 | 60 |
| Dry season | 92 | 85 | 42 | 65 |
| **2022/23** | **90** | **84** | **38** | **54** |
| Wet season | 88 | 80 | 36 | 48 |
| Dry season | 91 | 88 | 40 | 64 |
| **2021/23** | **93** | **85** | **40** | **59** |

ences in the statistical results of the study of Kalesse-Los et al. (2023). The object-based detection method revealed a 10 %-increase in warm cloud identification, and a 6 % decrease of trade wind cumuli occurrence of trade wind cumuli. Consequently, the proportion of virga originating from warm clouds using the object-based method is lower compared to the profile-based method, as more clouds are taken into account in the former. The proportion of precipitation-producing trade wind cumuli for the object-based detection method is notably lower compared to the profile-based detection method. This indicates that a large proportion of precipitation identified below trade wind cumuli for the profile-based method is counted as precipitation from warm clouds for the object-based detection method. The reason for these differences is related to the CBH configurations of the Virga-Sniffer and the different approaches to cloud classification. For the profile-based detection method, single profiles within a warm cloud object, identified by the object-based method, can be counted as trade wind cumuli by the profile-based method based on the location of the CBH below 1 km. These profiles with CBH lower than 1 km are simply omitted for the object-based method if they occur within a warm-cloud object.

The Virga-Sniffer tool is highly configurable and enables the replacement of the first CBH by the location of the LCL that can be estimated within the tool using surface measurements. Thus, the lowest CBH can be reassigned to the location of the LCL which can be observed in Fig. 4. The stratiform cloud shows CBHs above 1 km, indicated by the Cloudnet target classification product. In the used configurations (see Table A1), the Virga-Sniffer identifies this CBH at the location of the LCL, as the gap between the LCL and the true CBH is not sufficiently large enough (`cloud_max_gap`) to create a new CBH layer. We adjusted the Virga-Sniffer configurations for the RV *Meteor* to quantify the effect when the LCL is not used to replace the lowest CBH (configuration 2 in Table C1). We also reduced the CBH smoothing window to 15 s and set the CBH cleaning threshold to 1 %, which additionally results in more high clouds being included in the statistics. This reduces the proportions of warm clouds and trade wind cumuli for configuration 3 compared to configuration 2. Using these changes to the original Virga-Sniffer settings reduces the proportion of virga from trade wind cumuli to 24 % (when using the profile-based detection method, configuration 2) and thus makes the profile-based and object-based results more comparable.





**Table C1.** Virga-Sniffer statistics in % during the EUREC[4]A campaign between 18 January 2020 and 19 February 2020 over the BCO and the RV *Meteor*. The results are derived for different Virga-Sniffer configurations that are described in Table A1. Warm cloud and trade wind cumuli identification are performed using an object-based (ob) and a profile-based (pb) detection method. The statistics from the RV *Meteor*, using the profile-based approach and configuration 0 represent the results conducted by Kalesse-Los et al. (2023).

| Platform | BCO | RV *Meteor* | RV *Meteor* | RV *Meteor* | RV *Meteor* |
|---|---|---|---|---|---|
| Number of days: | 26 | 28 | 28 | 28 | 28 |
| N total cloud timesteps: | 34k | 34k | 358k | 411k | 484k |
| Configuration | 1 | 0 | 0 | 2 | 3 |
| Detection method | ob | ob | pb | pb | pb |
| **Warm clouds ...** | **71** | **81** | **73** | **79** | **73** |
| ... producing precipitation: | 47 | 44 | 56 | 51 | 50 |
| ... producing rain reaching the surface: | 5 | 10 | 14 | 12 | 10 |
| ... producing virga: | 42 | 34 | 42 | 39 | 40 |
| .. with haze echoes: (without/with virga in the same profile) | 43/60 | 7/11 | - | - | - |
| Fraction of virga from trade wind cumuli: | 70 | 31 | 56 | 48 | 24 |
| ... that are trade wind cumuli: | 78 | 49 | 63 | 59 | 40 |
| **Trade wind cumuli ...** | **56** | **40** | **46** | **46** | **29** |
| ... producing precipitation: | 44 | 29 | 59 | 49 | 37 |
| ... producing rain reaching the surface: | 6 | 7 | 22 | 17 | 13 |
| ... producing virga: | 38 | 22 | 37 | 32 | 24 |
| .. with haze echoes: (without/with virga in the same profile) | 49/70 | 10/12 | - | - | - |

## Appendix D: Seasonal comparison of the Virga-Sniffer statistics

A summary of the seasonal Virga-Sniffer statistics is provided in Table D1. The seasonal statistics form a possible reference value for future statistics that quantify the occurrence of virga.

*Code and data availability.* The source code of the haze echo classification (Roschke, 2024a, https://doi.org/10.5281/zenodo.10469906, v1.0.1) and cloud classification algorithm (Roschke, 2024b, https://doi.org/10.5281/zenodo.10471932, v1.0.1) is freely available and hosted on Zenodo. The BCO data are freely available to the broader community upon request. Cloudnet processing was done using CloudnetPy (Tukiainen et al., 2022, https://doi.org/10.5281/zenodo.7432587, v1.43.0).



**Table D1.** Virga-Sniffer Statistics in (%) for the dry and the wet season between July 2021 until July 2022 (2021/22) and July 2022 until July 2023 (2022/23). Precipitation and haze echoes are taken into account when CBH from warm clouds or trade wind cumuli are detected above. Haze echoes are filtered out when virga is detected in the same profile. The detection method object base (ob) is used for the BCO data set.

| period of observation | 07/2021 - 07/2022 | Wet season 2021/22 | Dry season 2021/22 | 07/2022 - 07/2023 | Wet season 2022/23 | Dry season 2022/23 | 07/2020 - 07/2023 |
|---|---|---|---|---|---|---|---|
| **Platform** | **BCO** | **BCO** | **BCO** | **BCO** | **BCO** | **BCO** | **BCO** |
| Number of days | 268 | 112 | 156 | 240 | 129 | 111 | 508 |
| Detection method | ob | ob | ob | ob | ob | ob | ob |
| Number of total cloud time steps | 397k | 159k | 238k | 342k | 184k | 158k | 739k |
| Fraction of clouds from clear sky | 51 | 49 | 53 | 50 | 49 | 50 | 50 |
| **Warm clouds ...** | **65** | **57** | **70** | **56** | **40** | **68** | **61** |
| ... producing precipitation | 51 | 47 | 54 | 43 | 40 | 45 | 48 |
| ... producing rain reaching the surface | 4 | 4 | 4 | 2 | 3 | 2 | 3 |
| ... producing virga | 47 | 43 | 50 | 41 | 37 | 43 | 44 |
| ... with haze echoes (without/ with virga in the same profile) | 40/65 | 46/69 | 37/63 | 43/62 | 42/63 | 43/64 | 41/64 |
| Fraction of virga from trade wind cumuli | 67 | 81 | 60 | 69 | 77 | 64 | 56 |
| ... that are trade wind cumuli | 74 | 86 | 68 | 75 | 81 | 70 | 68 |
| **Trade wind cumuli ...** | **48** | **49** | **48** | **42** | **37** | **47** | **45** |
| ... producing precipitation | 47 | 45 | 48 | 40 | 38 | 41 | 44 |
| ... producing rain reaching the surface | 4 | 4 | 4 | 2 | 3 | 2 | 3 |
| ... producing virga | 43 | 41 | 44 | 38 | 35 | 39 | 40 |
| ... with haze echoes | 50/74 | 51/74 | 49/74 | 50/68 | 47/63 | 53/68 | 50/71 |

*Author contributions.* This publication is based on the master thesis written by JR which was supervised by HKL and MH. JR developed the haze echo classification and cloud classification algorithm, performed the CloudnetPy processing and Virga-Sniffer processing of the BCO data, used the Virga-Sniffer output for analysis of the BCO and RV Meteor EUREC[4]A data, derived the statistics of the Cloudnet target classification data and contributed mostly to the writing of the manuscript. HKL was in charge of the project management and operation of the instruments of Leipzig University on board RV Meteor during EUREC[4]A. JW performed the Virga-Sniffer processing and statistics for additional configurations of the Virga-Sniffer for the EUREC[4]A dataset. HKL, MH, MK, JW, AF and AK contributed to discussions and editing of the manuscript. AK gave valuable feedback regarding the Virga-Sniffer statistics.

*Competing interests.* The contact author has declared that none of the authors has any competing interests.



*Acknowledgements.* The BCO data used in this publication were provided by the MPI for Meteorology Hamburg. We thank Lutz Hirsch, who provided scientific and technical support regarding the BCO data. The data are freely available to the broader community upon request. Special thanks to Johannes Bühl, Willi Schimmel and Teresa Vogl who gave technical support regarding the processing of the Cloudnet

dataset for the BCO. The RV *Meteor* data used in this publication were gathered in the EUREC[4]A field campaign and are made available through the AERIS portal by data upload through Leipzig University, MPI for Meteorology Hamburg, and the German Weather Service (DWD). EUREC[4]A is funded with the support of the European Research Council (ERC), the Max Planck Society (MPG), the German Research Foundation (DFG), the German Meteorological Service (DWD), and the German Aerospace Center (DLR). We acknowledge the pan-European Aerosol, Clouds, and Trace Gases research Infrastructure (ACTRIS) for providing the Cloudnet framework used in this study,

which was developed by the Finnish Meteorological Institute (FMI) and is available for download from https://cloudnet.fmi.fi/ (last access: 21 December 2021). We also acknowledge ECMWF for providing Integrated Forecasting System (IFS) model data as input for CloudnetPy. Parts of the results in this work make use of the color maps in the CMasher package (van der Velden, 2020). Special thanks to Nina Robbins and Ilya Serikov from MPI for Meteorology, Hamburg for discussions about the BCO Raman-lidar dataset collected during EUREC[4]A.

*Financial support.* This research has been supported by the European Social Fund (ESF) PV-WOW (grant no. 232101734). Further finan-

cial support was provided by the German Science Foundation (DFG; grant no. FO 1285/2-1). The paper was funded by the Open Access Publishing Fund of Leipzig University supported by the German Research Foundation within the program Open Access Publication Funding.



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
