# Peer review of "Discriminating between Drizzle or rain and sea salt aerosols in Cloudnet for measurements over the Barbados Cloud Observatory"

_EGUsphere, 2024_

## Author Response (AR1)

**Author's response to:**
**RC#1 from anonymous referee #1**
**https://doi.org/10.5194/egusphere-2024-894-RC1**

Johanna Roschke[1], Jonas Witthuhn[1,2], Marcus Klingebiel[1], Moritz Haarig[2], Andreas Foth[1], Anton Kötsche[1], and Heike Kalesse-Los[1]

[1]Leipzig Institute for Meteorology (LIM), Leipzig University, Leipzig, Germany
[2]Leibniz Institute for Tropospheric Research (TROPOS), Leipzig, Germany

**Correspondence:** johanna.roschke@uni-leipzig.de

Dear Referee #1,

Thank you for carefully reading the manuscript and pointing out several issues where the description needs to be improved for understanding. The requested clarifications and references to ambiguities contribute to the improvement of the manuscript.

In order to separate the reviewer's comments and the author's response, we printed the comments in black and the response in blue. Excerpts of the manuscript with marked changes are pinned directly to the appropriate responses, with the indicated text location (e.g., line number) referring to the manuscript in preprint.

Sincerely, on behalf of all authors

Johanna Roschke

**Changes to the Manuscript:**

We restructured the manuscript as suggested, moving the technical details of the Virga-Sniffer to a dedicated technical note in Roschke et al. (2024). This focuses the discussion on the introduced method and enhances clarity.

- Expanded the introduction with:

    - Importance of observing shallow cumulus convection and their macro- and microphysical properties.

    - Importance of sub-cloud evaporation in the trades.

    - Additional information about Cloudnet.

- Information in Sect. 2:

    - Added information about the Radar sensitivity and CFAD in Fig. 2

    - Included MWR and LWP processing in Sect. 2.3.

    - Improved Fig. 3 for better readability.

    - Moved Virga-Sniffer technical details (e.g., thresholds) to Sect. 2.4 of the technical note.

- Added information about the haze echo detection method in Sect. 3.1:

    - Added joint histograms for the radar reflectivity factor, mean Doppler velocity, and attenuated backscatter coefficient which show haze echo modes and explain the choice of parameters that determine the haze echo probability.

    - Added a table on how the choice of parameter influences combined haze echo probability.

    - Added Fig. 6 that illustrates the heuristic probability distributions for the radar reflectivity factor, mean Doppler velocity and attenuated backscatter coefficient.

- Cloud type classification in Sec. 3.2:

    - Clarified the reason of introducing cloud classification in Sect. 3.2.

    - Explained operational bounding box definition in Sect. 3.2.

- Restructured results in Sect. 4.3:

    - 4.3: Comparison to classifiers

    - 4.3.1: Comparison with the -50 dBZ threshold method

    - 4.3.2: Comparison with Virga-Sniffer

    - Added a case study (Fig. 11) that shows the comparison between different haze echo classifiers for a day where Drizzle or rain and haze echoes are detected simultaneously

– Added a case study (Fig. 12) exemplary for a day of low haze echo occurrence where further differences in the proportion of detected haze echoes between the classifiers become evident

– Added a long-term comparison of detected haze echo pixels for the classifiers (Fig.13)

– Updated Sect. 4.4 (limitations) with conclusions on shallow and deep cumulus radar reflectivity.

– Included Fig. 14 to illustrate a statistical comparison of radar reflectivities within shallow and deeper cumulus clouds for the case study presented in Fig. 11 and for the long-term period.

– Virga-Sniffer long-term statistics can now be found in the appendix

– Appendix:

– Virga-Sniffer configuration details can be found in the technical note.

Within the technical note for the Virga-Sniffer (Roschke et al., 2024), all datasets used in this study can be found. The technical note now contains the following information:

– description of configuration-specific thresholds that differ compared to the Virga-Sniffer configuration in Kalesse-Los et al. (2023).

– a section describing the CBH processing by the Virga-Sniffer, including a statistical comparison of CBH from the ceilometer internal cloud base detection algorithm and Cloudnet.

– an overview of different clutter-masks and velocity-masks and an explanation of how they influence virga and haze echo detection by the Virga-Sniffer.

– long-term statistical results by the Virga-Sniffer summarized in tables for different configurations

– analysis of object-based cloud classification statistics for the RV Meteor using the Virga-Sniffer configuration of Kalesse-Los et al. (2023)

**Response to RC#1 of Anonymous Referee #1:**

**General comments**

1. *GC1: The introduction summarizes that radar reflectivity thresholds are often used to exclude haze echoes from drizzle/rain occurrence analyses (L 71-73). The manuscript in its current state, however, does not clarify in what way the new haze category in Cloudnet improves or changes occurrence statistics compared to the Ze-threshold method (eg Klingebiel et al. (2019)). I would propose to add a comparison to the analysis Section (also see comment below).*

   * We agree, that the manuscript in its current state lacks a comparison between the new haze echo classification method and the traditional radar reflectivity threshold method. To address this, we added a comparison in the results section.

     > **Sect. 4.3.1 line 430.:**
     >
     > An analysis of the proportions of Cloudnet targets (including the new haze echo class) for all radar reflectivities below -50 dBZ for the period between 1 July 2021 to 1 July 2023 is shown in Fig. **??**. Fig. **??** a) shows that 12 % of all pixels with $Z_e < $ -50 dBZ are classified as "Liquid droplets", 25 % as "Drizzle or rain", 51 % as "Haze echoes" and 12 % as Insects and Aerosols. This means only about half of all targets that are filtered by the -50 dBZ-threshold are haze echoes. Applying a -50 dBZ-threshold-based method would lead to an underestimation of 6 % of all pixels classified as "Drizzle or rain" and an underestimation of 4 % of all pixels classified as "Liquid droplets". While this proportion seems rather small, it can influence precipitation statistics for specific case studies.

2. *GC2: The authors need to clarify where to find the data sets that they used for their analysis. Many datasets, especially those obtained during the EUREC4A period, are publically available with dois; were these data sets used?*

   * We agree that data should be publicly available to enhance transparency and reproducibility in research. The BCO instrument data are available upon request from MPI Hamburg, while all datasets used in our study (including Cloudnet and Virga-Sniffer data) are publicly accessible in the technical note under zenodo in Roschke et al. (2024)

3. *GC3: Sec 3.1, L247 - 259: I am missing the reasons for the choice of μ, sigma, beta, and a justification for how they were optimized and set. The authors should clarify how sensitive the analyses are to these settings including the probability threshold of 60% (L261). How do users of the method need to adjust these settings for different maritime Cloudnet sites (or is it "plug and play"?)?*

   * We agree that the current version of the manuscript misses the explanation for the choice of parameter values that defines the haze echo probability. We added a statistic and a sensitivity study to the respective section.

4. *GC4: I find the argumentation line of the analysis at times hard to follow. In order to be more convincing of the new method, I would propose to adapt the structure as follows:*

- – *4 i) Apply method to BCO statistics and evaluate new approach by comparing to Virga-Sniffer and traditional -50dBZ threshold in order to highlight benefits of new classification scheme for haze detection; include an analysis of limitations using spectra/skewness (also see Specific comments below) and sensitivity to parameters (see comment above)*

- – *4 ii) analyze driving factors of haze occurrences in water vapor or subsidence space (see comment below)*

- – *(optional:) 4iii) make use of Cloudnet and Virga-Sniffer to analyze virga and precip statistics at BCO given the improved detection scheme excluding mis-labeled haze)*

\* We agree that, due to the complexity of the haze echo classification method and of the Virga-Sniffer tool, the current manuscript structure could be clearer and more streamlined to effectively communicate the benefits of the new haze echo classification scheme. However, we note that radar Doppler spectra and skewness data were not available in the BCO dataset available to us, which limits our ability to perform a full analysis of these aspects as suggested. We have moved the detailed Virga-Sniffer results and configuration information to a supplementary Zenodo publication in Roschke et al. (2024). This allows us to streamline the manuscript and focus on the key findings.

5. *GC5: the main message of the paper should be clarified throughout the manuscript. Is the main scope of the manuscript to introduce a new Cloudnet classification scheme? Or, rather, to analyze rain and virga characteristics at BCO given an optimized detection method? The abstract and introduction rather focus on the novel Cloudnet method, while the main scope of the analysis Section seems to focus on BCO statistics.*

\* We understand that the main message may seem split between the introduction of the new Cloudnet classification scheme and the comparisons of precipitation statistics. To clarify, this work has two goals: first, to introduce and evaluate the Cloudnet classification for haze echo and drizzle detection, and second, to use the improved Cloudnet target classification to provide valuable, long-term statistical insight into cloud and precipitation characteristics at BCO. Until now, long-term statistics of Cloudnet classification targets at the BCO are not published, so we believe that the inclusion of these statistics adds significant value and relevance to the manuscript.

**Specific comments**

- – *L 21-26: The importance of evaporation and moistening processes in the sub-cloud layer for cloud and precipitation evolution should be highlighted here.*

\* We added this information to the introduction.
* * *
**Sect. 1 line 28.:**

Precipitation from trade wind cumuli often occurs in the form of drizzle (Wu et al., 2017) —that often evaporates before reaching the ground Kalesse-Los et al. (2023). Precipitation evaporation influences the moisture and heat budgets of clouds themselves (Emanuel et al., 1994) as well as the subcloud environment via the formation of cold pools (Langhans and Romps, 2015).
* * *
– *L24: A reference should be added.*

\* We added references to the repetive line.

> **Sect. 1 line 28.**:
>
> Their spatial structure and evolution can be influenced by precipitation (Albrecht et al., 1995; Albrecht, 1993).

– *L177: To my knowledge, operation of the CORAL Radar and ceilometer continued at BCO after the EUREC4A campaign (http://bcoweb.mpimet.mpg.de/systems/data_availability/DeviceAvailability.html, last access July 2, 24)*

\* Indeed, the CORAL radar and ceilometer continued measuring at the BCO after the EUREC4A campaign. However, we would like to clarify that no Liquid Water Path (LWP) data are available because the microwave radiometer only resumed operation in July 2021. Consequently, Cloudnet processing could not be performed due to the lack of continuous LWP data. We clarified this in the manuscript.

> **Sect. 2.3 line 188.**:
>
>  The BCOHAT was not operating at the BCO after the EUREC$^4$A campaign in February 2020 until July  2021, which is why Cloudnet data could not be retrieved.

– *L178: the authors should clarify why this data is not usable and why timestamps cannot be corrected in post-processing.*

\* We agree that it's important to clarify why certain data were not usable. To date, five ceilometers have operated at the BCO between 2010 and 2020. Problems like dusty sensors or calibration factors are not reported in detail. This makes it challenging to process Cloudnet data for different ceilometers. Between 2011 and 2015, we encountered significant issues with the ceilometer data during Cloudnet processing due to random time jumps in the recorded timestamps. These time shifts can occur when the laser quality is low or when the radome is not clean. This makes the ceilometer system reboot itself, which in turn can lead to time stamp jumps. However, we do not know if this is the actual reason for these time jumps. The time shifts are not consistent or systematic, making it extremely difficult to identify a clear pattern for correction. Given the randomness of these timestamp jumps, it remains uncertain whether the timestamps are duplicated at actual true time steps or if the entire time axis is shifted forward or backward. This lack of clarity complicates efforts to adjust timestamps reliably in post-processing, as we cannot determine whether to correct timestamps into the future or the past. Importantly, these time jumps did not occur every day but sporadically, leading to gaps in the data. Moreover, a low laser quality makes it difficult to interpret the attenuated backscatter signal within our method. In conclusion, to maintain data integrity and avoid the potential influence of calibration differences across multiple ceilometers, we have chosen to focus our analysis on two years of data from after 2021, during which no time jumps were observed.

– *Sec 2.2.1 and Sec. 2.2.2; L 569: it remains unclear to me throughout the manuscript how MWR and MRR data impact the classification algorithm. Are they mandatory for the new classification class? The HATPRO in operation at the time*

*of analysis is the BCOHAT instrument as specified in Schnitt et al, 2024, ESSD (doi.org/10.5194/essd-16-681-2024) (reference missing)*

* We acknowledge that the manuscript could clarify how these datasets influence the classification process and add information accordingly. Additionally, we appreciate your point regarding the HATPRO instrument. As a note, it is important to mention that in the metadata of the LWP files, the instrument source is listed as "RPG-HATPRO-G2" rather than BCOHAT. Moreover, the retrieval of the LWP for the MWR is provided by the RPG Radiometer Physics GmbH and was retrieved by a neural network.

> **Sect. 2.3 line 135-140.:**
>
>  The Humidity and temperature profiling radiometer (BCOHAT) measures seven brightness temperatures around the water vapor absorption band between $22 - 31\,\text{GHz}$ and in the oxygen absorption complex between $51 - 58\,\text{GHz}$. Measurements around the water vapor absorption line are used to derive a column-integrated liquid water path (LWP) which is retrieved by a neural network provided by the RPG Radiometer physics GmbH. The vertical resolution is less than $40\,\text{m}$ in the sub-cloud layer with a temporal resolution of $4\,\text{s}$. Data from the current microwave radiometer are available since April 2017 (Stevens et al., 2016).

– *Fig 3: I would suggest to add boxes in (a) which illustrate the zoomed areas in (b) and (c); and to maybe re-configure the plot such that (a) is largest on the left side, and (b) and (c) are smaller and connected to boxes in (a)*

* We appreciate your insight on how to improve the clarity and presentation of the figure. We improved the representation of the figure which can be found in Roschke et al. (2024)

– *Fig 4 : in order to highlight the added value of the new classification class compared to the conventional Ze threshold method (L70), I would propose to add a panel on the top to illustrate the measured radar reflectivity.*

* We understand the importance of highlighting the added value of the new classification class compared to the conventional Ze threshold method. We have added a comparison including a case studies to the result section. There the added value becomes more clear.

– *L205: arguments for why m and c were chosen this way should be added here. How sensitive is the clutter mask and resulting analysis and evaluation to these values? I suspect that the presented evaluation of the Cloudnet class with the Virga-Sniffer strongly depends on the values chosen here.*

* To provide clarity, we have moved the Virga-Sniffer configuration section, along with a more detailed sensitivity analysis regarding the clutter mask, to the Zenodo publication in Roschke et al. (2024). In the manuscript, we note that changing the clutter mask to $m = -55$ and $c = -38$ significantly reduces the proportion of identified virga by the Virga-Sniffer for classified haze echo pixels to $0\,\%$. In this scenario, the proportion of haze echoes identified using our developed method, compared to the Virga-Sniffer, would increase to $89\,\%$.

– *L205: do the authors refer to the sensitivity limit of the CORAL radar? If so, this limit should scale with range, and should be negative. If not, a clarification is needed here.*

\* We agree that the sensitivity limit should scale with range. We added a joined histogram to the instrument section (also see comment of Rev2.). However, for the Virga-Sniffer clutter-mask, the minimum reflectivity is used to derive a linear function that filters clutter. We oriented our configurations to the original paper of the Virga-Sniffer, where the clutter-mask does not scale with range.

– *L232: a sentence should be added on how the insect detection scheme works and why it would be suitable for also detecting haze; this information should also be added to Sec 2.3. Why not including the Virga-Sniffer method to Cloudnet instead or in addition as it uses similar instruments? The advantages of the chosen method compared to the Virga-Sniffer should be highlighted.*

\* We appreciate your suggestion for clarification. The method we employ for insect detection is indeed similar to the approach used in Cloudnet. Insects are classified by combining heuristic probabilities derived from various radar parameters, along with additional variables such as temperature. However, it is important to note that the insect detection method is novel and still requires validation, as highlighted in the Cloudnetpy code. While the Virga-Sniffer is highly configurable, it operates as an independent tool and is not part of the Cloudnet framework. The advantages of our approach include its integration within the Cloudnet target classification scheme, allowing for reconfigurability to marine Cloudnet sites and their specific instrumentation. Additionally, our haze echo detection method incorporates the ceilometer, which is primarily used in the Virga-Sniffer for cloud base height identification. This added instrument enhances our ability to detect haze echoes more effectively. We will ensure to add a brief explanation of how the insect detection scheme operates and its suitability for haze detection in Section 3 of the manuscript.

> **Sect. 3 line 229-234.**:
>
> This section gives an overview of the method that was developed to discriminate between sea salt aerosols and "Drizzle or rain" in Cloudnet. The method is similar to the approach for insect detection in Cloudnet. Insects are classified by combining the heuristic probabilities derived from various radar parameters and additional variables such as temperature. As highlighted in the CloudnetPy code, insect detection is novel and still needs to be validated. The advantage of using a similar approach is, that it can be easily implemented within the Cloudnet target classification scheme and that it is configurable for marine Cloudnet sites and their particular instrumentation.

– *L290: doubles L261.*

\* Thank you for your observation regarding the duplicated sentence in Line 290. We have removed the redundant sentence.

– *L 311: more explanation is needed for why it would be important and interesting to split the analysis in the two cloud classes; this should be stated already in the introduction as well.*

* We added an explanation to the following section. Also see answer to Rev.2.

> **Sect. 3.2 line 314-317.:**
>
> Existing statistics on clouds an precipitation over Barbados focus on warm clouds and trade wind cumuli (Kalesse-Los et al., 2023; Nuijens et al., 2014; Acquistapace et al., 2019; Schulz et al., 2021). In order to compare our statistics with existing literature and to investigate precipitation properties of warm clouds and haze echo occurrence an object-based cloud classifier was developed.

– *Sec 4.1: Rather than splitting the analysis into dry and wet season statistics for each year, an occurrence analysis could be performed in subsidence or water vapor space for both years to exclude for example skewing wet intrusions in the dry season from the statistics. The EUREC4A period could be used to analyze driving factors in more detail, such as cloud organization type, cloud type, wind direction, wind speed, and to include the impact of Saharan dust events on haze occurrence (which is mentioned in L403 but not shown).*

* Thank you for your thoughtful suggestions regarding the analysis in Section 4.1. We appreciate your perspective on performing an occurrence analysis in subsidence or water vapor space for both years. Our analysis is oriented towards established statistics, such as those from Stevens et al. (2016); Nuijens et al. (2014) and Nuijens et al. (2015), which focus specifically on dry and wet season statistics. We recognize the significance of analyzing driving factors like cloud organization type, wind direction, and the impact of Saharan dust events on haze occurrence. However, it's important to note that the primary focus of this study is to introduce a haze echo classification method in Cloudnet and its validation, while further statistics on driving factors could be interesting for future studies.

– *Sec 4.3: The authors should clarify why they are comparing BCO and the Meteor observations; I am confused - did the authors also run Cloudnet based on the Meteor? If so, additional input is needed in Sec 2 and the introduction. Maybe the authors rather use the comparison to optimize the application of the Virga-Sniffer to BCO measurements in which case the text needs to be clarified to underline this.*

* Thank you for highlighting the need for clarification regarding the comparison between BCO and RV Meteor observations in Section 4.3. We conducted the Virga-Sniffer comparison during the EUREC$^4$A campaign to evaluate the configuration differences between measurements from the RV *Meteor* and BCO during the same period. This approach allows us to explore how measurement variations arise due to differing instruments, particularly radar wavelength and ensures consistency by examining a common period. While this is partially noted in Line 407, we have now expanded this explanation to clearly state that the goal was not to run Cloudnet specifically on the RV *Meteor* but rather to assess the implications of using different instrument configurations on virga and haze echo detection during an overlapping observation period.

> **Appendix 1 line 572-577.:**
>
> This comparison contextualizes the long-term virga statistics at the BCO by aligning them with the published results of Kalesse-Los et al. (2023). By comparing data from both platforms during the same time period, we aim to highlight differences due to instrument variations and provide a more comprehensive understanding of the statistics across different observation environments.

– *L425 and Fig 9: The text should comment on the large occurrence of 'Unclassified' aboard the Meteor compared to the BCO; and should summarize why the difference between object- and profile-based statistics is particularly profound for the trade wind cumulus class in panel (b) compared to the warm clouds class in panel (a) (which is hinted at in L434, and extensively analysed in Appendix C, but should be summarized here)*

\* The category "unclassified" in this context represents time steps where clouds are observed but are not precipitating. We updated the figure legend and clarified this in the text.

– *Sec 4.3.2 As I understand the analysis presented here, the Virga-Sniffer is applied to BCO measurements and occurrence statistics of virga, precipitation and clouds are analysed. I am not sure how this Section relates to the title of the manuscript, as the Cloudnet haze method is not included in this Section (also see GC 4 and 5)*

\* Thank you for your feedback regarding Section 4.3.2 and its relation to the title of the manuscript. We agree that the analysis presented in this section, which focuses on the application of the Virga-Sniffer to BCO measurements and the occurrence statistics of virga, precipitation, and clouds, may not align directly with the main focus of the paper on the Cloudnet haze classification method. Consequently, we included a comparison of our method to other classifiers such as the Virga-Sniffer and the -50 dBZ threshold method. Virga-Sniffer-related statistics can now be found in the appendix or the zenodo technical note of Roschke et al. (2024)

– *L476: it should be clarified if Virga-Sniffer results are shown, or Cloudnet classification results; also see comment above*

\* See comment above.

– *L502: Spectra and higher moments are available at MPI for the analyzed period and should be used in the analysis to strengthen the proposed method, or, at least, to quantify the limitations more thoroughly. Could the classification scheme be adapted to include skewness as an additional proxy for detecting haze?*

\* Thank you for your suggestion to incorporate spectra and higher moments, particularly skewness, as additional proxies for detecting haze. We understand that these metrics could potentially strengthen the proposed method. However, as noted in the manuscript, neither spectral data nor skewness was available in the input files accessible to us for the analyzed period. Skewness could indeed be added as an additional input parameter in future analyses if this data becomes available. Incorporating a skewness probability into the combined haze echo probability, could enhance the robustness of the classification scheme.

**Technical Corrections**

- *Fig 2: All colors seem to be related to 24h data coverage; the colorbar should be adjusted to enhance the Figure's message.*

- *Fig 8: Legend should be adjusted to distinguish solid and dashed line without reading the caption.*

- *Fig 9 caption: last sentence should be moved to main body text.*

- *Figs 7-10: description of colors shown in legends need to be added to the captions.*

- Expanded the introduction with:

    - Importance of observing shallow cumulus convection and their macro- and microphysical properties.

    - Importance of sub-cloud evaporation in the trades.

    - Additional information about Cloudnet.

- Information in Sect. 2:

    - Added information about the Radar sensitivity and CFAD in Fig. 2

    - Included MWR and LWP processing in Sect. 2.3.

    - Improved Fig. 3 for better readability.

    - Moved Virga-Sniffer technical details (e.g., thresholds) to Sect. 2.4 of the technical note.

- Added information about the haze echo detection method in Sect. 3.1:

    - Added joint histograms for the radar reflectivity factor, mean Doppler velocity, and attenuated backscatter coefficient which show haze echo modes and explain the choice of parameters that determine the haze echo probability.

    - Added a table on how the choice of parameter influences combined haze echo probability.

    - Added Fig. 6 that illustrates the heuristic probability distributions for the radar reflectivity factor, mean Doppler velocity and attenuated backscatter coefficient.

- Cloud type classification in Sec. 3.2:

    - Clarified the reason of introducing cloud classification in Sect. 3.2.

    - Explained operational bounding box definition in Sect. 3.2.

- Restructured results in Sect. 4.3:

    - 4.3: Comparison to classifiers

    - 4.3.1: Comparison with the -50 dBZ threshold method

    - 4.3.2: Comparison with Virga-Sniffer

    - Added a case study (Fig. 11) that shows the comparison between different haze echo classifiers for a day where Drizzle or rain and haze echoes are detected simultaneously

– Added a case study (Fig. 12) exemplary for a day of low haze echo occurrence where further differences in the proportion of detected haze echoes between the classifiers become evident

– Added a long-term comparison of detected haze echo pixels for the classifiers (Fig.13)

– Updated Sect. 4.4 (limitations) with conclusions on shallow and deep cumulus radar reflectivity.

– Included Fig. 14 to illustrate a statistical comparison of radar reflectivities within shallow and deeper cumulus clouds for the case study presented in Fig. 11 and for the long-term period.

– Virga-Sniffer long-term statistics can now be found in the appendix

– Appendix:

– Virga-Sniffer configuration details can be found in the technical note.

Within the technical note for the Virga-Sniffer (Roschke et al., 2024), all datasets used in this study can be found. The technical note now contains the following information:

– description of configuration-specific thresholds that differ compared to the Virga-Sniffer configuration in Kalesse-Los et al. (2023).

– a section describing the CBH processing by the Virga-Sniffer, including a statistical comparison of CBH from the ceilometer internal cloud base detection algorithm and Cloudnet.

– an overview of different clutter-masks and velocity-masks and an explanation of how they influence virga and haze echo detection by the Virga-Sniffer.

– long-term statistical results by the Virga-Sniffer summarized in tables for different configurations

– analysis of object-based cloud classification statistics for the RV Meteor using the Virga-Sniffer configuration of Kalesse-Los et al. (2023)

**Response to RC#2 of Anonymous Referee #3:**

TODO: Answer the comments, stick to the format: -review comment, -author answer, latexdiff of section in question

**General comment**

*The paper develops a new method to detect haze echoes in Cloudnet and it is a potentially powerful tool to be deployed in all Cloudnet stations over the ocean to reduce biases and errors in virga and precipitation detection. Quantifying correctly virga and precipitation over the ocean is a relevant scientific question and fits, in my opinion, within the scope of AMT. Moreover, I can envision useful applications in model evaluation. The algorithm is based on previous research methods and the presented approach is quite solid, verified on a long dataset from the Barbados Cloud Observatory. I generally like the approach, the language is fluent, formulas are well written, but I would recommend publishing after major revisions, which I suggest because I found that the methods could be more clearly outlined to better guide the reader in the technicalities of the approach. I think that some choices of parameters need a more solid justification and maybe some results need to be stated in a stronger outstanding way.*

TODO:answer text Based on the suggestions of two reviewers, we have substantially revised the manuscript.

**Major comments**

1. *The paper aims to introduce a new method, but besides that, it provides a lot of detailed analysis of the virga sniffer method, which is the subject of a former publication by Kalesse-Los et al.. Especially in the appendix, many sections are given to provide insights on the virga sniffer, which is not the core of this publication. In this respect I see two possibilities: either include the new method as part of the virga sniffer method (a sort of part 2 of the former paper) or remove most of the virga sniffer-related analysis. I am saying this because it is confusing to follow the argumentation among the different methods used, and in the end, it is not clear what is the best configuration to use.*

   * We agree and moved all Virga-Sniffer results that do not contain haze echo statistics to the technical note publication of this study. In Roschke et al. (2024) we describe the Virga-Sniffer configurations for the BCO.

2. *I think that you need to explain and justify the values you choose for the radar reflectivity, mean Doppler velocity, and ceilometer backscatter haze distributions that you use to calculate your probabilities, I could not find it. Moreover, I don't understand why, in the example case study of Figure 5, the method identifies haze echoes only below shallow cumulus clouds. In my understanding, the identification should be independent from the cloud presence, so I would love to see more case studies, and see details of the case shown as well. See the comment on the plot for some suggested analysis. I understand that in part of the paper, you then decide to focus on the detection of haze frequency for the different cloud types, but in general, why restrict to profiles where there is a cloud base only? and if this is not what you are doing, please go through the doubts and the descriptions because it is not understandable what you are doing.*

* We agree that the current version of the manuscript misses the explanation for the choice of parameter values that defines the haze echo probability. We added a statistic and a sensitivity study to the methodology in Sect.3.

3. *The paper is full of very detailed discussions where the reader can easily get lost. My suggestion is to go over the comments and think if all the details are really needed and possibly reduce some of them for the sake of readability and understanding. I know that the authors who created the algorithms think that details are necessary, but for a reader who is not familiar with the algorithm, they are totally overwhelming.*

* Thank you for the suggestion, we substantially restructured the manuscript. See comment 1

4. *I would also suggest reconsidering the title after having tackled the major comment number 1.*

* After restructuring the mauscript we decide to keep the title.

**Specific comments**

1. *line 30: what is the minimum sensitivity of the radar at BCO? Can you refer to a CFAD Ze vs Height plot for the BCO site from some papers or add one? That would be nice.*

* We confirm that the minimum radar reflectivity of the CORAL cloud radar for a 10 s time resolution is -70 dBZ at a height of 500 m, as described in the Section 2. We corrected, that for a 2 s measurement configuration, the radars sensitivity has decreased to -62 dBZ at an altitude of 500 m and -41 dBZ at an altitude of 5 km. In response to your comment, we have added a joint histogram of the radar reflectivity factor with height to Section 2 and added a note in the instrument section.

> **Sect. 2.1 line 115-121.**:
>
> Compared to the 10  s resolution, the 2 s measurement configuration of the radar has led to a sensitivity decrease to -62 dBZ at an altitude of 500 m and -41 dBZ at an altitude of 5 km. The joint histogram of radar reflectivity per range can be seen Fig. **??**). The histogram is normalized by the total number of counts per radar range gate, such that the histogram values represent the frequency of occurrence. There seems to be a cutoff at around 300 m for reflectivities below -60 dBZ which is related to radar antenna near field effects. The calibration of the radar shows an uncertainty of 1.3 dB for the radar reflectivity measurements (Görsdorf et al., 2015).

2. *line 57 (highlighted sentence): I would formulate as follows: The size at which cloud droplets transition to precipitation varies in the literature*

* Thank you for your suggestion. We have revised the highlighted sentence in line 57

> **Sect. 1 line 59.:**
>
> The size  at which cloud droplets transition to precipitation varies in the literature.

3. *line 54: Is this drizzle definition a definition you created or something from literature? please specify.*

* We have clarified in the manuscript that the definition is based on established literature sources.

> **Sect. 1 line 60.:**
>
> Glienke et al. (2017) defines drizzle as drops that are large enough to have fall velocities that exceed the typical fluctuations of vertical velocity in the cloud. For a reasonable range of stratocumulus vertical velocities of 0.1 to $1\,\mathrm{ms}^{-1}$, the corresponding diameters are approximately 50 to $250\,\mu m$ (Glienke et al., 2017).

4. *line 76: Maybe here you can add also that Cloudnet is used on multiple sites in the world, especially the target classification allows to compare statistical cloud properties in a homogeneous way from different sites, just to give value to the Cloudnet tool.*

* We agree and have now included additional context about Cloudnet. However, in section 2.5 we already state that with Cloudnet uniform data sets are created and can be used to evaluate cloud profiles between various measurement stations.

> **Sect. 1 line 81-87.:**
>
> Synergistic retrievals such as Cloudnet, provide the potential to identify different hydrometeors by applying state-of-the-art data processing chains for a complex combination of data from ground-based remote sensing instruments (Illingworth et al., 2007). Cloudnet, which began in 2002 as a European research project with three stations, has since developed into a continuously operating network of 25 stations throughout Europe. Currently, Cloudnet is funded by the European Commission under the Seventh Framework Program as part of ACTRIS (Aerosol, Clouds and Trace Gases Research Infrastructure) (Laj et al., 2024). Cloudnet offers a range of products, including the target classification scheme, designed to identify the physical phase of hydrometeors.

5. *line 112: Is it possible to add a Cfad of Ze for the whole statistic of the data you use, instead of only mentioning the variation of the sensitivity with height? see also comment 1.*

* See comment 1.

6. *line 131: which retrieval? maybe cite what is used at BCO to retrieve LWP and IWV.*

* We acknowledge that the manuscript could clarify how these datasets influence the classification process and add information accordingly. Additionally, we appreciate your point regarding the HATPRO instrument. As a note, it is important

to mention that in the metadata of the LWP files, the instrument source is listed as "RPG-HATPRO-G2" rather than BCOHAT. Moreover, the retrieval of the LWP for the MWR is provided by the RPG Radiometer Physics GmbH and was retrieved by a neural network. .
* * *
**Sect. 2.4 line 135.:**

**0.1   Microwave radiometer**

The Humidity and temperature profiling radiometer (BCOHAT) measures seven brightness temperatures around the water vapor absorption band between $22-31$ GHz and in the oxygen absorption complex between $51-58$ GHz. Measurements around the water vapor absorption line are used to derive a column-integrated liquid water path (LWP) which is retrieved by a neural network provided by the RPG Radiometer physics GmbH.
* * *
7. *line 143: I tend to disagree. A Cloudnet station, in my experience, should include the 3 instruments needed for cloud profiling, i.e. cloud radar, aerosol lidar, and microwave radiometer, https://www.actris.eu/facilities/national-facilities/observational-platforms . Moreover, in this document, also a disdrometer is included, for a site to be a CCRES (Actris center for cloud remote sensing) https://www.actris.eu/sites/default/files/CCRES/CCRES%20Requirements%2010112022.docx.pdf*

\* We have adjusted the text to clarify the requirements for a Cloudnet station accordingly.
* * *
**Sect. 2.5 line 150.:**

The basic instrumentation of a Cloudnet station includes a cloud radar, a ceilometer and a MWR. Additional instruments are a  MRR, a rain gauge or a distrometer. Information on the rain rate from the MRR is used to flag time steps where rain is reaching the ground. The MWR is needed to provide liquid water path (LWP) which is used to correct for liquid attenuation in the cloud radar measurements within the CloudnetPy processing. The observations from the instruments are combined with thermodynamic data from a model or radiosonde to accurately characterize clouds up to 15 km with high temporal and vertical resolution (Illingworth et al., 2007). In this study, Cloudnet data is processed with CloudnetPy (Tukiainen et al., 2020, version 1.43.1).
* * *
8. *line 145: I would not define the MWR as an optional instrument, since it also provides IWV and LWP with lower uncertainty compared to the single channel retrieval which has to deal with the unknown impact of water vapor on the LWP estimation. Moreover, it has scanning options that allow monitoring of temperature and humidity fields around the site, which can become more and more relevant for model evaluation in the future.*

* Thank you for highlighting the importance of the microwave radiometer (MWR) for Cloudnet. We have revised the text accordingly (also see comment above)

9. *line 165: What is the pixel size of Cloudnet? maybe it would be good to mention the range resolution and the time resolution which is also the one of your output.*

* We agree that mentioning the range and time resolution earlier in the manuscript will provide more clarity for readers.

> **Sect. 2.3 line 178-189.**:
> ______________________________
>
> Within CloudnetPy data from different instruments are interpolated onto a common grid with a temporal resolution of 30 s and height resolution of 30m (Illingworth et al., 2007). In the Cloudnet target classification (Hogan and O'Connor, 2004), grid points [...] . The interested reader is referred to the original paper by Hogan and O'Connor (2004) for additional information about the Cloudnet target classification procedure.

10. *line 175; I would put the reference at the beginning of the paragraph " In the Cloudnet target classification (Hogan and O'Connor, 2004).*

* We have moved the reference to the beginning of the paragraph on line 175, following the suggested placement (see comment above).

11. *line 188: why not use the same algorithm from (Tuononen et al., 2019) you mentioned before? is it a problem of resolution? How do they compare with the other ones?*

* The Virga-Sniffer does not calculate CBH from the attenuated backscatter coefficient but instead uses CBH as an input. The CBH is further processed by the Virga-Sniffer to fill gaps and classify virga for all profiles within the cloud. This CBH processing involves several configuration parameters that must be chosen in a balanced manner. Consequently, the CBH processing in the Virga-Sniffer is fundamentally different from the algorithm of Tuononen et al. (2019). To enhance the input for the Virga-Sniffer, we merged CBH data from the ceilometer with Cloudnet data. This merging increases the number of CBHs provided as input to the Virga-Sniffer. We observed that incorporating additional CBH data from the ceilometer leads to an increase in the number of cold cloud profiles in the final Virga-Sniffer output. Capturing the maximum number of CBHs is essential for the virga statistics (per time step), as these show the proportion of warm clouds or trade wind cumuli relative to the total number of detected clouds. We acknowledge that this distinction was not clearly explained in the original text, and we have revised the manuscript accordingly.

> **Sect. 2.6 line 200-203.**:
> ______________________________
>
>  is described in Roschke et al. (2024).

12. *line 190: what do you mean by "by more than the set of thresholds?" Can't you just say more generally " In multi-layer cloud situations CBH is assigned with a more complex processing (see Paper, appendix)? also, do we need to know in this paper the details of the multilayer cloud-based detection of virga sniffers? I would just cite the Virga sniffer paper.*

* We agree and moved the explanation of the BCO Virga-Sniffer configuration to the technical note in Roschke et al. (2024). There we state more clearly how to adjust the Virga-Sniffer for multilayer cloud situations.

13. *line 201: why not a surface disdrometer? or rain gauge? isn't it available at BCO? At what height is the lowest valid MRR range gate used?*

* The MRR is continuously operating at the BCO and the disdrometer was not measuring in 2022. Consequently, we only included the rain rate of the MRR as an input for the Cloudnet processing. The lowest MRR range gate is $125\,\mathrm{m}$.

14. *line 203: Is this corrected for air motion? might be tricky to use the mean Doppler velocity as a proxy for hydrometeor fall speed.*

* It is indeed tricky to use the mean Doppler velocity as a proxy for hydrometeor fall speed. Consequently, in Virga-Sniffer configuration for the BCO only drizzle in downdrafts is considered as in the study of Kalesse-Los et al. (2023). Drizzle drops with terminal fall velocity comparable to the expected updrafts, remain unclassified in this study. Correcting the mean Doppler velocity with the vertical velocity measured by a Doppler lidar would improve the efficiency of the algorithm's ability of separating virga from haze echoes. This can be incorporated in future studies as well as into the Cloudnet target classification scheme for haze echo detection. Moreover, Doppler lidar measurements will be harmonized for ACTRIS Cloudnet stations in the future. In case the BCO becomes a Cloudnet station, Doppler lidar data could be integrated into the Cloudnet target classification to improve haze echo detection. We added a comment on this topic to the technical note in Roschke et al. (2024)

15. *formula 1: from my understanding, this is the equation of the line that defines the clutter mask. if it is so, please state it clearly. I would find it easier to follow if you introduce the different masks that appear in the plot (vm mask and clutter mask) and then define how you identify the virga and the haze echoes given such masks. Virga mask appears at the beginning and then not anymore. it is a bit confusing to follow also because it is not clear the goal, which I think is to distinguish haze from virga. or? I hope these questions can help improve the clarity.*

* We restructured the text regarding the virga-mask and clutter-mask and moved these technical descriptions to the technical note in Roschke et al. (2024). In addition, we stated more clearly how the Virga-Sniffer identifies haze echoes.

> **Sect. 2.6 line 211-217.:**
>
>  with radar reflectivity factor below -50 dBZ *that are not classified as virga by the Virga-Sniffer. Unclassified radar signals can represent cloud, precipitation, and haze echo pixels when no information about the CBH is available. Pixels not classified by the Virga-Sniffer while information about CBH is available and rain is not detected at the surface are filtered by the clutter- and velocity-mask (Kalesse-Los et al., 2023). As a novelty, in this study, the clutter-mask was modified so, that all* unclassified radar signals with radar  *reflectivities* below -50  dBZ  *are identified as haze echoes by* the Virga-Sniffer *. The configurations of both the clutter-mask and velocity mask are detailed in Roschke et al. (2024).*

16. *line 205: how do you determine m and c values? this is explained in the virga sniffer paper (eq 1 in par 3.4 of that paper) even if I did not find in that paragraph how you came up with the values of m and c. However, why repeat it entirely again here? Can't you make here a shorter and simpler summary, citing the paper published for details? I find this virga sniffer description very long and distracting from the topic of the paper, which is the other algorithm for sea salt /drizzle discrimination.*

   * We moved the repective section to the zenodo publication in Roschke et al. (2024). Also see comment above.

17. *line 205: at which height? considering all heights? not clear*

   * The clutter-mask and velocity-mask are pixel based. Consequently, there is no height dependence and the mask are not scaled by range. Also see comment of Rev.1.

18. *line 207: Is filtering out sea salt something you also do in Virga sniffer? or is this something you do for determining the prob thresholds you use in the alg of this paper? if it is the second, I would put it in the section of this new algorithm, with a subsection called "threshold determination". My main problem is that throughout the whole discussion, It is never very clear what parts are in the virga sniffer and what belongs only to the new algorithm.*

   * Haze echoes can be identified using the Virga-Sniffer (see comment above). Due to the new structure of the manuscript, we hope that the discussion is now clearer. We introduce the new section: "Comparison to classifiers" where we compare the haze echo detection of the new method to the Virga-Sniffer haze echo detection and the -50 dBZ-threshold approach that is commonly used in many BCO studies.

19. *line 209: are clutter c and clutter m the parameters c and m in eq 1? If yes, use the same formalism. If not, explain and explain also if they are linked. Still, unclear how you determine the actual values of the parameters.*

   * TODO:We adjusted the text in the technical note of Roschke et al. (2024) accordingly.

20. *line 223: but why do so if the cloud base in Cloudnet (Tuononen et al,. 2019) performs better? at least this is what is evident from your selected case study.*

* TODO: The differences in the CBH in the case study are related to the Virga-Sniffer CBH processing. We added an explanation to the technical note in Roschke et al. (2024)

21. *line 232: reference to the insect detection method missing*

   * See answer to review 1

> **Sect. 3 line 230-234.**:
> ___
> This section gives an overview of the method that was developed to discriminate between sea salt aerosols and "Drizzle or rain" in Cloudnet. The method is similar to the approach for insect detection in Cloudnet. Insects are classified by combining the heuristic probabilities derived from various radar parameters and additional variables such as temperature. As highlighted in the CloudnetPy code, insect detection is novel and still needs to be validated. The advantage of using a similar approach is, that it can be easily implemented within the Cloudnet target classification scheme and that it is configurable for marine Cloudnet sites and their particular instrumentation.

22. *figure 4: I don't understand the discrepancy in the cloud base from Cloudnet and the virga sniffer at 6:30, in precipitation conditions. To me it looks like the cloud base detected from Cloudnet is more reliable, based on Tuononen et al, 2019) so why not use it? I understand that it is on 30 s resolution and probably the Virga sniffer is a higher resolution, but this can be interpolated. The LCL base causes a big bias in the virga depth values you obtain. Or am I misunderstanding?*

   * The LCL is not used to replace the Cloudnet CBH in the Virga-Sniffer processing. The reason for the bias in the virga depth values arise due to additional Virga-Sniffer processing steps. We added an explanaition to the technical note in Roschke et al. (2024).

23. *line 245. and around: In my understanding, the probabilities should depend on the parameters of the distributions of your observables for haze. Now, can we see what your distributions for Ze, Vd, and Beta look like for haze and also for the other hydrometeors, if possible? can you somehow justify your choices of the parameters mu and sigma? otherwise, we just have to believe. It would be also nice to see the same distributions for the Cloudnet classes, to understand how much different they are.*

   * We have added two figures to the manuscript in section 3 that show the haze echo modes in the observations for $Z_e$, $V_d$, and $\beta$ as well their heuristic probability distributions and a table that illustrates the choice of the probability threshold. Please note that the individual Cloudnet target classifications (Cloud droplets only, Drizzle or rain, ...) are based on a profile-based approach, and not on the probability approach presented here. For further details, we refer to the original paper by Hogan and O'Connor (2004).

24. *line 248: how did you decide such values?*

\* The parameters controlling the behavior of the probability distributions are derived from statistical analysis. We have included 2D histograms in FIg.5 illustrating the radar and ceilometer measurements to exemplify our choice of parameters. Also see comment above.
* * *
**Sect. 3.1 line 235-255.:**

 The frequent occurrence of haze echoes at the BCO becomes evident in the 2D histogram in Fig. 1 a) for the radar reflectivity factor and the mean Doppler velocity for the period between 1 July 2021 and 1 July 2022. Two distinct modes with high data point density are visible. The first mode is centered around a radar reflectivity factor of -60 dBZ and a mean Doppler velocity of $0.2 \, \mathrm{m\,s^{-1}}$, which we attribute to the frequent occurrence of haze echoes over the BCO. The second mode, represents the cloud and precipitation mode with a high data point density at radar reflectivities from -40 dBZ to 20 dBZ. The haze echo mode is also evident in the 2D histogram of the attenuated backscatter coefficient and mean Doppler velocity is shown in Fig.,1 b). The first mode is centered at approximately $0.2 \, \mathrm{m\,s^{-1}}$ and spans a broad range of attenuated backscatter coefficients, from about $0.3 \times 10^{-6} \, \mathrm{m^{-1}\,sr^{-1}}$ to $1.3 \times 10^{-6} \, \mathrm{m^{-1}\,sr^{-1}}$. In contrast, the second mode peaks at $-0.5 \, \mathrm{m\,s^{-1}}$ and is associated with attenuated backscatter coefficients below $0.5 \times 10^{-6} \, \mathrm{m^{-1}\,sr^{-1}}$. The first mode is attributed to haze echoes, while the second likely corresponds to pixels characterized by lower aerosol loads, potentially caused by wet deposition due to precipitation.The two modes are distinctly separated in the 2D histogram of radar reflectivity factor and attenuated backscatter coefficient (Fig.,1 c)). The first mode is associated with radar reflectivities below -50 dBZ and attenuated backscatter coefficients ranging between $0.3 \times 10^{-6} \, \mathrm{m^{-1}\,sr^{-1}}$ and $1.3 \times 10^{-6} \, \mathrm{m^{-1}\,sr^{-1}}$. The second mode, with radar reflectivities between -50 and -20 dBZ, corresponds to attenuated backscatter coefficients below $0.5 \times 10^{-6} \, \mathrm{m^{-1}\,sr^{-1}}$. Once again, the first mode is attributed to haze echoes, whereas the second mode is linked to clouds or precipitating particles.Following these observations, we isolate the haze echo mode by deriving a combined haze echo probability from heuristic probability functions from individual parameters (namely radar reflectivity, radar mean Doppler velocity and ceilometer attenuated backscatter coefficient). For each pixel, a probability  is estimated.

[Figure]

**Figure 1.** BCO 35 GHz cloud radar and ceilometer (1 July 2021 – 1 July 2022): Histograms of the mean Doppler velocity and radar reflectivity factor in (a), the attenuated backscatter coefficient and mean Doppler velocity in (b) and the attenuated backscatter coefficient and radar reflectivity factor in (c). Radar reflectivity bin width is $1\,\mathrm{dBZ}$, the Doppler velocity bin width $0.1\,\mathrm{m\,s^{-1}}$ and the attenuated backscatter coefficient bin width is $2.3{\times}10^{-6}\,\mathrm{m^{-1}\,sr^{-1}}$. Note that due to the attenuation of the ceilometer signal within liquid layers, the number of data points is lower in (b) and (c). Histogram area for a minimum number of 500 data points that fulfill haze echo condition 1 (Tab. 1) are marked by the red line.

25. *lines 250 and 257: same question as before, how did you decide such values? maybe you can plot some distributions for all variables to show where are the values coming from? I am not sure I understand otherwise.*

* We have extended our description and included a Figure that showcases the distributions for all variables. Also see answer to comment 23.
* * *
**Sect. 3.1 line 275-291.:**

By performing element-wise multiplication of the haze echo probability arrays for radar reflectivity factor, mean Doppler velocity and ceilometer attenuated backscatter coefficient, the combined probability ($P_{combined}$) can be estimated. When the  $P_{combined}$ exceeds 60 % for grid points below the CBH or at altitudes below 2 km (average top of the height of the marine aerosol layer over Barbados)  in cloud-free situations, the haze echo category is implemented and replaces targets previously classified as "Drizzle or rain" in the Cloudnet target classification. The heuristic probability distributions for radar and ceilometer variables are visualized in Fig.2 for the combination of probabilities in line 1 (condition 1) of Table 1. In scenarios where the probability of the attenuated backscatter coefficient ($P_{\beta}$) reaches 70 %, the probability of the mean Doppler velocity ($P_v$) must be close to 100 % when the probability of the radar reflectivity factor ($Z_e$) is 86 %, to reach a combined probability threshold of ($P_{combined}$) greater than 60 % for haze echo identification. $P_{\beta}$ reaches 70 % for $\beta$ values between $0.3 \times 10^{-6}$ m$^{-1}$ sr$^{-1}$ and $1.3 \times 10^{-6}$ m$^{-1}$ sr$^{-1}$. Minimum and maximum values for each variable's probability are summarized in Tab. 1. For the example scenario of $P_{Z_e}$ =86 %, the mean Doppler velocity must be greater than -0.36 m s$^{-1}$ for maximum radar reflectivity factors of -50 dBZ. For stronger downdrafts, with velocities down to -0.78 m s$^{-1}$, the radar reflectivity factor needs to be lower than -60.46 dBZ for haze echoes to be classified. Note, that if any individual probability is at 60 %, the remaining probabilities must reach 100 % for haze echo classification. In such cases, the minimum mean Doppler velocity is -0.95 m s$^{-1}$, and the maximum radar reflectivity factor is -46.46 dBZ (see Tab. 1).
* * *
[Figure]

**Figure 2.** Heuristic probability distribution for the radar reflectivity factor ($P_{Ze}$) in a), mean Doppler velocity ($P_v$) in b), and attenuated backscatter coefficient ($P_\beta$) in c) together with the selected values of the parameters $\mu$, $\sigma$, and $\beta$ for each distribution. The respective values can be also found in Tab.1.

**Table 1.** Minimum probabilities for the radar reflectivity factor ($P_{Ze}$), mean Doppler velocity ($P_v$), and attenuated backscatter coefficient ($P_\beta$), along with their respective minimum and maximum values required to achieve a combined probability ($P_{combined}$) of 60 %.

| | Probabilities | | | | Variable Values | | | |
|---|---|---|---|---|---|---|---|---|
| Convention | $P_{Ze}$ | $P_v$ | $P_\beta$ | $P_{combined}$ | $Z_{e,\max}$ | $v_{\min}$ | $\beta_{\min}$ | $\beta_{\max}$ |
| Units | % | % | % | % | dBZ | $\mathrm{m\,s^{-1}}$ | $\mathrm{sr^{-1}m^{-1}}$ | $\mathrm{sr^{-1}m^{-1}}$ |
| Max. $Z_e$ for $P_\beta = 70\%$ (Figure 1 condition 1, Figure 2) | $> 86$ | 100 | $> 70$ | 60 | $< -50.38$ | $> -0.36$ | $> 0.4 \times 10^{-6}$ | $< 1.06 \times 10^{-6}$ |
| Min. $v$ for $P_\beta = 70\%$ | 100 | $> 86$ | $> 70$ | 60 | $< -60.46$ | $> -0.78$ | $> 0.4 \times 10^{-6}$ | $< 1.06 \times 10^{-6}$ |
| Min./Max. $\beta$ | 100 | 100 | 60 | 60 | $< -60.46$ | $> -0.36$ | $> 0.3 \times 10^{-6}$ | $< 1.1 \times 10^{-6}$ |
| Max. $Z_e$ | 60 | 100 | 100 | 60 | $< -46.28$ | $> -0.36$ | $> 0.56 \times 10^{-6}$ | $< 0.84 \times 10^{-6}$ |
| Min. $v$ | 100 | 60 | 100 | 60 | $< -60.46$ | $> -0.95$ | $> 0.56 \times 10^{-6}$ | $< 0.84 \times 10^{-6}$ |

26. *line 260: how do you define haze distributions for Ze, Vm, and Beta? I assume you define haze using the masks, but then it would be nice to see the distributions of all the variables, not just the case study.*

   * see comments 23 and 24.

27. *line 264: From my understanding, the classification is pixel based. I think it would be something to highlight. Also, this probability does not depend on the cloud base, so potentially you can classify as haze pixels that do not have a cloud base above. So why is the plot of Figure 5 haze is found only below the cloud base? is this common in other case studies too?*

   * The classification is indeed pixel-based, and we will highlight this aspect in the text. While the probability does not depend on the cloud base, we restrict the application of the haze echo identification method to pixels that are classified as drizzle or rain. This approach helps to ensure that cloud pixels are not mistakenly identified as haze echoes. However, it is important to note that haze echoes can occur when no CBH is deteced, as illustrated in our case study at 07:00 UTC. Without the haze echo identification, these pixels would have been classified by Cloudnet solely as Drizzle or rain, even in the absence of clouds above. We further included an analysis that reveals the occurrence of haze echoes in the measurements at the BCO

28. *figure 5: it would be nice to have a zoomed plot of the area from 7:00 to 8:00. My concern is this: if there is sea salt spray and the environment has humidity for sea salt condensation nuclei to grow, why do you see it only below the small shallow clouds? in theory you should see it on all timestamps in the sub cloud layer, independently of the cloud presence above, at least assuming that the humidity is homogeneous. Do you have a humidity profiler to display humidity profile time series and understand if, for some reason below the cloud base, there's a higher amount of humidity or some temperature variations making water vapor condensation easier? maybe displaying also IWV time series from the MWR can help, or some T, RH time series from the surface station in the worst case? Maybe you have an explanation to understand why you have such gaps? might be that my thinking makes no sense for some reasons.*

* In general sea salt aerosols are present and visible in the radar measurements also when no CBH is detected. This can be seen in Fig.3 for the 23 January 2023 BCO case study below. The relative humidity (RH) is not that homogeneous as the model or a single radiosonde suggest. In the case of updrafts, RH increases and often a cloud is formed. Whereas in downdraft regions, RH is lower and there is no cloud. Consequently, grown sea salt particles are more prone to be found below a cloud in an updraft region. Nevertheless, they can be present independently of the cloud, e.g., if we are in an updraft region but the cloud was not formed yet. In other words, the cloud and the grown sea salt are caused by the same updraft. Measurements of RH over the BCO are available from the ECMWF model but for the higher temporal resolution which is needed here, it has to be calculated using the water vapor mixing ratio from the CORAL Raman lidar. The calculation of RH using the water vapor mixing ratio are performed using temperature and pressure data from either a model or a radiosounding. The nearest radiosounding is from the airport in Barbados (78954 TBPB Grantley Adams Observations), approximately 13 km south of Deeples Point. However, the radiosounding only provides observations at 0 and 12 UTC, which makes interpolation onto the Cloudnet resolution not accurate due to the limited number of observations (00 UTC, 12 UTC, and 00 UTC of the next day) per day. The vertical profiles of the CORAL lidar observations on 23 January 2023 between 00:45 and 01:45 UTC are shown in Fig.4. The observations reveal a two-layer structure with the marine boundary layer (MBL) extending up to 1 km with RH up to approximately 100 % and the marine aerosol layer (MAL) between 1 and 2 km with lower RH compared to the MBL. Consequently, sea salt aerosols that exist in the MAL, are smaller, as they can not take up as much water as in the MBL. This might explain, why they are not visible in the radar measurements above approximately 0.8 km. The hygroscopic growth can be observed in Fig.5 for the same case study up to 2.2 km. This proves, that sea salt aerosols are present in the atmospheric column above 0.8 km. As RH increase the particle linear depolarization ratio (PLDR) decreases, as sea salt particles take up water and become more spherical in shape as has been intensivly studied by Haarig et al. (2017) at Barbados. At the same time, the backscatter derived Ångström exponent decreases which indicates that particles increase in size. This is further supported by values of the radar reflectivity factor that increase as the Ångström exponent decreases. We did not include the Raman lidar and the RH measurements in the manuscript for several reasons: radiosoundings from the airport might not be representative of the variations in the MBL, given the limited number of measurements available. Moreover, the RH from the ECMWF model exhibits large differences compared to the radiosounding at times close to the observations

(Fig.6). Furthermore, consistent and reliable profiles of the water vapor mixing ratio from Raman lidar data are rare and mostly available during nighttime, and radiosounding data might not always be available.

[Figure]

**Figure 3.** Case Study 2: BCO on 23 January 2023, 01:15-01:45 UTC. Cloudnet target classification in (a), radar reflectivity factor in (b) and Doppler velocity in (c). The LCL is marked as a dotted line.

[Figure]

**Figure 4.** Marine aerosol layer over the BCO on 23 January 2023, 01:15-01:45 UTC. Relative humidity (derived from CORAL lidar water vapor mixing ratio together with pressure and temperature data from the ECMWF model) in a), particle linear depolarization ratio (PLDR) in b), and particle backscatter coefficient at two different wavelengths (355 and 532 nm) in c)-d). The lidar data were interpolated onto the Cloudnet grid (30 s temporal and 30 m vertical resolution). The dotted line marks the LCL.

[Figure]

**Figure 5.** Lidar variables recorded over the BCO on 23 January 2023 at 01:15-01:45 UTC. Correlation of the particle linear depolarization ratio (at 355 nm) (a), particle backscatter coefficient (at 355 and 532 nm (b-c) and backscatter Ångström exponent (d) with RH (derived from lidar (CORAL) water vapor mixing ratio and temperature and pressure data from the ECMWF model). The data was averaged over 5 min and has a vertical resolution of 30 m. For pixels with valid radar reflectivities, the lidar variables are color-coded.

[Figure]

**Figure 6.** One-hour mean profiles of relative humidity (RH) in (a) and water vapor mixing ratio in (b) at the BCO between 00:00-01:00UTC on 23 January, 2023. The RH was estimated using temperature and pressure data from the ECMWF model and the radiosounding (RS) data from the nearby airport (78954 TBPB Grantley Adams Observations), combined with the water vapor mixing ratio data from the CORAL Raman lidar. The profiles of RH for the ECMWF model and the radiosounding are depicted by darker lines in comparison to the combined measurements.

29.  *line 291: I see that in the final plot, you used the cloud base from Cloudnet to detect virga around 6:30 and not the one from the virga sniffer tool (Fig 4b) So I did not fully get at the end which is the cloud base you use in virga sniffer from the description of the virga identification. Maybe you can improve the description?*

 \* To answer this question, we have added the section "Cloud base height detection" in the technical note in Roschke et al. (2024) of the Virga-Sniffer configuration for the BCO. Also see other answers on Virga-Sniffer CBH related comments.

30. *Section 3.2: I think that at the beginning of this paragraph, you need to explain that you want to study haze occurrence under a cloud base and investigate its frequencies for different types of clouds. This is why, I think, you introduce the cloud type classification out of the blue at the moment, which is not needed otherwise given that your haze identification is pixel-based depending on the probability assigned to a tuple of $Z_e$, $v_d$, and $\beta$.*

 \* The reason for the introduction of different cloud classes is that we want to compare the final statistics of clouds and precipitation to existing studies. We added an introductory statement to the corresponding section. We admit that in the section "Limitations," we discuss the occurrence of haze echoes below different cloud types, and this might lead to confusion regarding the reason for introducing the cloud classification. To clarify our point in the "Limitations" section: The question "Drizzle or haze echoes?" can only be asked when there is a cloud present above (without considering the signals caused by tilted precipitation fall streaks). Consequently, we analyze how often haze echoes occur below shallow and deeper cumulus clouds to identify how often a misclassification of haze echoes for true drizzle signals by our method might occur.

> **Sect. 3.2 line 322-325.:**
> ───────────────────────
> Existing statistics on clouds and precipitation over Barbados focus on warm clouds and trade wind cumuli , e.g., (Kalesse-Los et al., 2023; Nuijens et al., 2014; Acquistapace et al., 2019; Schulz et al., 2021). In order to compare our statistics with existing literature and to investigate precipitation properties of warm clouds and haze echo occurrence, an object-based cloud classifier was developed.

31. *line 310: Maybe you can have a subsection "object-based cloud classification". Somewhere before you also need to introduce the profile-based classification method, which is? Is it the Virga sniffer? it is unclear to me what it is that you later call the profile-based method. Please clarify it here, before using them in the results, and explain why you introduced them before, as suggested in the previous comment. My impression is that a good position would be at the beginning of this section 3.2. For both methods, please introduce the acronyms you use later here.*

 \* We agree that introducing both the object-based and profile-based classification methods earlier in Section 3.2 will improve clarity.

> **Sect. 3.2 line 325-334.**:
>
> Object-based method in this context stands for an approach to identify continuous radar signals in time and space (connected pixels). Once a cloud object has been detected, information about the CBH can be analyzed within this object. In  contrast, profile-based cloud identification relies on the number of detected cloud base heights (CBH). Profile-based approaches like Cloudnet or the Virga-Sniffer suffer from two issues: Firstly, if gaps occur in the CBH measurements of the ceilometer (e.g., for multilayer cloud situations or during heavy precipitation), even when the radar detects signals from clouds, profile-based methods do not register these cloud profiles. Secondly, when distinguishing different clouds by the altitude of their CBH, profile-based approaches could erroneously classify individual cloud profiles within a single cloud object as different cloud types if the CBH varies strongly within the cloud.

32. *line 332: how do you operationally define the bounding box?*

  * The bounding box is operationally defined for each hydrometeor cluster by the lowest and uppermost pixels at a specific height, as well as the first and last pixels at a specific time stamp. This is automatically calculated using the regionprops function from the scikit-image library. We will update the manuscript to clarify this definition.

> **Sect. X line 334-350.**:
>
> The object-based cloud classification algorithm, utilizing object-based feature detection methods from the scikit-image library(Van der Walt et al., 2014), is applied to all radar signals, excluding haze echo radar pixels. As a result, haze echo occurrence statistics are unaffected by the cloud classification, as haze echoe pixels are counted independently whenever they occur in a profile. A hydrometeor cluster is defined as a group of at least three connected pixels. CTH is determined by the highest located radar pixel of a hydrometeor cluster that can be estimated from the location of the bounding box (Fig. **??**) of the cluster. The bounding box bounds for each cluster are extracted using the `regionprops` function from the `skimage.measure` module, which is part of the scikit-image library (Van der Walt et al., 2014).

33. *line 370: what is the link between haze echo occurrences and cloud base height? not very clear (see also points above)*

* In response to your comment, we have restructured the relevant section to provide a clearer explanation.

> **Sect. 3.2 line 344-347.:**
>
> The object-based cloud classification algorithm, utilizing object-based feature detection methods from the scikit-image library(Van der Walt et al., 2014), is applied to all radar signals, excluding haze echo radar pixels. As a result, haze echo occurrence statistics are unaffected by the cloud classification, as haze echoe pixels are counted independently whenever they occur in a profile.

34. *section name 4.3: I would call it "Method validation using Virga sniffer". It would be nice to find a name for the method presented in the paper..*

   * We prefer not to give the method a specific name, as the goal is to integrate it into the CloudnetPy processing framework rather than treating it as an independent tool. The primary focus is on utilizing Cloudnet's target categorization and classification products for the identification process, which makes it more of an integrated approach, rather than a standalone method.

35.  *section 4.3: It is very confusing for the reader to understand an evaluation that is based on different cloud definitions, and different methods to identify clouds. Is it so crucial to add here and not in the appendix the comparison with ship obs? I think that by presenting the evaluation only on BCO data would give value to the analysis of the long-term statistics, without putting too many things to see on the plate. I would compare with Meteor in a different section, maybe in the appendix. It is very difficult to follow in this way because of all these differences, which for you are obvious because you created the algorithms, but can make readers get lost and miss the point.*

   * We understand that the presentation of different cloud definitions and methods for identifying clouds may be confusing for the reader. To address this concern, we have moved all "standalone" Virga-Sniffer results including the RV Meteor comparison to the appendix.

36. *line 413: I think you need to mention how Virga sniffers detect precip and haze briefly: profile method based on the thresholds...or whatever it is, specify, concisely.*

* We agree that the detection method for precipitating and haze echoes needs to be mentioned more clearly.
* * *
**Sect. 2.6 line 204-208.:**

The detection of precipitation, clouds, and CTH is performed by analyzing radar signals. Precipitation is identified at each range gate of the radar reflectivity mask below CBH. The process involves downward assignment from the CBH until the lowest radar signal. Surface rain is detected from radar reflectivities in combination with the Cloudnet rain flag, which incorporates measurements  from the MRR. The virga-mask is  refined through the incorporation of cloud radar mean Doppler velocity data.  The velocity-mask restricts virga from occurring for negative mean Doppler velocities. [..] . The configurations of both the clutter-mask and velocity mask are detailed inRoschke et al. (2024).
* * *
37. *section 4.3.1: What is the main message of par 4.3.1? I don't find a clear message from this section, because all results seem to be conditioned by some aspects and don't provide a more general conclusion. I would suggest sharpening the core message without specifying so many details and moving all that is not the core message in the appendix. It is very dispersive for the reader. I understand that you created such algorithms and you are interested in their main differences, but out of those details, what is that we are learning about the processes? What is the statistical info you want modelers to remember when they read the paper to find results to compare their runs with (I am making an example, here, of possible users of your work)*

* We understand that Section 4.3.1 currently lacks a clear core message, with many results being conditioned by specific aspects. We hope that the new structure of the manuscript will sharpen the core message.

38. *fig 9: How does it compare with old Cloudnet? is there a way to show that, instead of only showing Virga sniffer results? I think a good message is also to show the improvement with what is the current state of the art, so the standard Cloudnet categories of drizzle/rain.*

* This is included already in the Fig.10, where it can be seen in c) that the occurrence of haze echoes leads to an overestimation of 16 % of Drizzle or rain in the dry season within the "old" Cloudnet target classification

39. *line 448: it is quite dispersive so it is not easy to remember the difference between the two methods. I would recall here that the developed haze method is based on probability and the Virga sniffer haze echo detection is based on ?? I understood a combination of thresholds, but I am not sure, and add that you want to compare them.*

* We agree that the Virga-Sniffer haze echo detection could be recalled before comparing them to the introduced method.

> **Sect. 4.3 line 424-427.:**
> ───────────────────────────────
> This subsection compares the haze echo detection method with the -50 dBZ-reflectivity threshold method and the results from the Virga-Sniffer. Recall that the Virga-Sniffer haze echo detection (cf. Sect. **??**) uses a profile-based cloud identification method combined with a pixel-based precipitation detection while the newly developed haze echo detection method is solely pixel-based.

40. *line 464: Can you please motivate again this choice? (also commented before) I don't understand why consider haze only when there's a cloud above. I don't understand why you want to exclude the cases that occur without a cloud base, as if they are physically different. In my view, they should not be different, and they should not be excluded.*

* See answer to comment 30. The respective section can now be found in the appendix.

41. *line 476: what do you mean here exactly by rain proportion? Have you checked the rain amounts? or is it a count of rain occurrence? please clarify..*

* The proportion of rain refers to the amount of pixels related to the occurrence of rain at the surface, which can be identified using the Virga-Sniffer rain flag. This process is already described in the Virga-Sniffer method section.

42. *line 505: where? refer to the plot, would be nice to follow your words with visual support*

* We added case studies to section 4.3 where the simultaneous occurrence of haze echoes and Drizzle or rain can be observed.

[Figure]

**Figure 7.** BCO case study 19 December 2022: Cloudnet target classification (old) a), Cloudnet target classification (with new haze echo category) b), Virga-Sniffer output c), radar reflectivity factor d), radar mean Doppler velocity e) and ceilometer attenuated backscatter coefficient f). Pixels with radar reflectivity below -50 dBZ are hatched in b) and c).

43.  *line 506: I think this is a too strong sentence if you want to base it on results from Acquistapace et al., 2019's case study. In Fig 13b of Acquistapace et al. 2019, it is clearly visible that the distribution of drizzle mature, which is the larger drizzle stage, does not have a sharp peak at -10 dBz, but instead, it shows a tail extending to -40. Therefore, it is not true that clouds with Ze lower than -10 dBZ do not indicate the presence of drizzle. Your values below -10 dBZ completely match the drizzle growth stage, which has a long tail of values of mean Doppler velocities corresponding to falling hydrometeors (fig. 13d).*

* TODO:We agree and have adjusted the text accordingly.

> **Sect. X line Xff.:**
> ___________
>
> As shown in Fig. ??a), for shallow cumulus clouds, the in-cloud radar reflectivity factor remains well below -10 dBZ  , both for the December 19 2022 case study and the long-term period. Both distributions could indicate the presence of  larger drizzle drops. However, we can not be certain because of lacking skewness information. For the long-term statistics, only 17 % of  radar reflectivity values within shallow cumulus clouds range between -30 and -10 dBZ.  , indicating possible drizzle growth and reaching the typical drizzle thresholds as suggested by Kogan et al. (2012) and Frisch et al. (1995). This low percentage implies that, while drizzle formation inside the clouds is possible, it is not dominant in the shallow cumulus clouds  at the BCO. A study by Albright et al. (2023) at the BCO further supports the idea that very shallow clouds over Barbados rarely produce precipitation. From their observations, they hypothesize that a large part of the condensate from clouds, that form within the transition layer between 550 and 700 m and have CTH below 1.3 km, evaporates as the role of these clouds is to humidify the transition layer. In the two-year dataset analyzed here, we also detect haze echoes below these very shallow clouds and  in line with Albright et al. (2023) conclude that for these clouds, misclassification of precipitation as sea salt aerosols is unlikely.

44.  *line 541: do you have an example of this situation to discuss with plots?*

* It remains unclear to us, what this question refers to.

45.  *line 589: Why is this section in this paper and not in the Virga sniffer paper? Do we need it if in the end you used the cloud base from Cloudnet (fig 5) and you seem to suggest that the cloud base detection algorithm in Tuononen et al, 2019 is performing better? I am asking for the sake of the readability. In the paper, it is a continuous jump between the virga sniffer and the algorithm you want to present in this paper, and it is very dispersive. ( see comments above)*

* We agree that the transitions are somewhat disjointed, and we have moved the Virga Sniffer content to the appendix and the technical note publication.

46. *line 589: designed*

   * Thank you for noting the word repetition; we have revised the text accordingly.

47.  *appendix c: Again: are you presenting the virga sniffer tool or the other algorithm? I find it confusing to present an evaluation of an algorithm in your previous paper in the paper in which you introduce a new algorithm. I suggest removing it and publishing it elsewhere. Or, just change the title and structure and introduce this new algorithm as a development of Virga sniffer. It is extremely hard to follow for me, maybe it is my problem.*

   * We agree thus have substantially revised the manuscript.

48. *table B1: is this stat relevant to the sea salt discrimination that you have in the title? I think it is off-topic.*

   * Please refer to the general comment for further details.

**References**

[revised manuscript text omitted]